# Conformal Language Generation with Collaborative Ranking and Dynamic Thresholds

## Abstract

Large language models (LLMs) face significant challenges in providing reliable uncertainty quantification for language generation. We introduce a novel conformal prediction framework specifically designed to enhance this reliability through Collaborative Ranking and Dynamic Thresholds. Our method innovatively departs from traditional metrics by harnessing advanced LLM capabilities for comparative judgment, allowing it to rank candidate responses and form a robust, rank-based nonconformity score. This approach enables the construction of prediction sets with rigorous statistical guarantees that inherently adapt to diverse input difficulties and prompt complexities. Extensive experiments across varied question-answering domains consistently demonstrate significant improvements in conditional coverage, delivering precisely calibrated LLM outputs demanding extended reasoning and factual accuracy. We have provided code with implementation details in the repository below: `https://anonymous.4open.science/r/512499`.

## 1 Introduction

Large Language Models (LLMs) generate human-like text across diverse tasks but often lack reliable uncertainty quantification, leading to hallucinations (Ji et al., 2023; Huang et al., 2025a). This issue is critical in high-stakes domains like healthcare or education, where factual accuracy is paramount (Maynez et al., 2020; Tang et al., 2023).

Various approaches have been developed to quantify uncertainty in LLM outputs, including probability-based thresholds for sentence-level calibration (Desai & Durrett, 2020; Huang et al., 2025b), token-level early stopping (Glushkova et al., 2021; Mohri & Hashimoto, 2024), and LLM self-evaluation (Kadavath et al., 2022b; Lin et al., 2022). However, these methods typically lack formal statistical guarantees and struggle with consistency across different input types.

Conformal prediction (CP) (Vovk et al., 2005; Shafer & Vovk, 2008; et al., 2014; Luo & Zhou, 2026; 2025) offers a robust framework for providing statistical guarantees on model outputs without strong modeling assumptions. It transforms predictions from any black-box model into valid prediction sets, guaranteed to contain the true outcome with high probability. Recent work has applied CP to LLMs for multiple-choice question answering (Kumar et al., 2023), token-level predictions (Ravfogel et al., 2023), and confidence sets for open-ended generation (Quach et al., 2024). Mohri and Hashimoto (Mohri & Hashimoto, 2024) notably introduced conformal factuality, using entailment sets to dynamically adjust LLM responses while maintaining trustworthiness.

Despite these advances, existing CP methods for LLMs face significant limitations: they often provide only marginal guarantees, failing to account for varying input difficulty (Cherian et al., 2024; Vovk, 2012); employ inefficient filtering due to weakly correlated scoring functions (Mohri & Hashimoto, 2024); and frequently violate the exchangeability assumption (Wang et al., 2025b).

To address these, we propose a novel collaborative ranking conformal method. This approach uses a lower-version LLM to generate multiple candidate answers, which a higher-version model then ranks by quality and factual accuracy. Conformal prediction is applied to the selected answer, establishing statistical guarantees. This rank-based mechanism provides dynamic, instance-specific thresholds, offering a more nuanced quality assessment than confidence scores and enhancing uncertainty quantification.

Our main contributions are as follows:

- We propose a **ranking-based** scoring function specifically designed for LLMs. This model overcomes the limitations of traditional probabilistic metrics by **collaboratively generating response-evaluation rankings**.

- Our approach adjusts the rank adaptively to the input difficulty, which enables instance-specific ranking thresholds that dynamically respond to query difficulty, significantly **renhancing conditional coverage** across diverse question types.

- We demonstrate through experiments on complex question-answering tasks that our approach achieves superior performance compared to existing methods.

The remainder of this paper presents related work (Section 2), preliminaries and problem setup (Section 3), details our rank-based conformal prediction methodology and explains how we enhance conditional validity through difficulty-adaptive thresholds (Section 4). We then introduce the experimental design (Section 5) and evaluate our method on multiple question-answering benchmark datasets (Section 6). Finally, we supplement the Appendix with ablation studies on individual parameters(Appendix B), prompt design, and a specific implementation case(Appendix C).

## 2 RELATED WORK

### 2.1 CONFORMAL PREDICTION FOR LARGE LANGUAGE MODELS

Conformal Prediction (CP) Vovk et al. (2005) offers a distribution-free, model-agnostic framework for statistically guaranteed prediction sets. Split CP (SCP) Hebiri (2010) simplifies this by dividing data into calibration and test sets, suitable for modern machine learning.

Given LLM issues like hallucinations Ji et al. (2022), poor calibration Desai & Durrett (2020); Kong et al. (2020), and biases Gallegos et al. (2023); Guo et al. (2022), reliable uncertainty quantification is vital Min et al. (2023b). CP provides a principled solution with theoretical coverage guarantees Angelopoulos & Bates (2022).

In question answering, Kumar et al. (2023) applied SCP to multiple-choice tasks, extended to open-ended generation (white-box and black-box) by Quach et al. (2024); Wang et al. (2025b). Mohri & Hashimoto (2024) introduced "conformal factuality" to filter invalid LLM claims. For sequence generation, Deutschmann et al. (2024) extended beam search with CP for guaranteed sequence sets, while Su et al. (2024) quantified LM uncertainty without logit access.

The combinatorial complexity of autoregressive text generation poses unique CP challenges. Ravfogel et al. (2023) addressed overconfidence with conformal nucleus sampling and adaptive prediction sets. Ulmer et al. (2024) extended this using non-exchangeable CP Barber et al. (2023) and k-nearest neighbors in hidden state space. Yu et al. (2023) also developed coverage guarantees for beam search despite intractable sequence space.

### 2.2 ENHANCED CONDITIONAL VALIDITY GUARANTEES

Traditional CP offers only marginal guarantees, often insufficient for specific inputs or groups. Gibbs et al. (2025) introduced conditional CP to approximate guarantees for specified function classes. Other work focused on group conditional guarantees Vovk (2012); Toccaceli & Gammerman (2019); Gupta et al. (2020); Ding et al. (2023); Dunn et al. (2023); Kiyani et al. (2024), including Mondrian CP for disjoint groups Vovk et al. (2003). Romano et al. (2020) achieved equitable coverage for disjoint protected groups, and Foygel Barber et al. (2021) proposed a computationally intensive method for overlapping groups. Jung et al. (2023) enhanced conditional coverage using quantile regression with subgroup indicators, albeit with distributional assumptions.

Cherian et al. (2024) extended conditional guarantees to language models via level-adaptive CP, employing "conditional boosting" and "level-adaptive prediction." Wang et al. (2025b)'s SConU improved cross-domain guarantees by filtering uncertainty outliers, addressing exchangeability. For multimodal LLMs, Wang et al. (2025a) developed TRON, a two-step framework for calibrating response requirements and applying nonconformity scores for risk-controlled, high-quality outputs.

The efficacy of conformal methods depends on scoring functions. Stutz et al. (2022) automated score improvement via differentiation through the split conformal algorithm. Kiyani et al. (2024) reframed

score optimization as a min-max task for optimal LM conformal scores. These techniques enhance practical utility, ensuring valid and informative prediction sets.

## 3 PRELIMINARIES

### 3.1 PROBLEM SETUP

We begin by formalizing the problem of uncertainty quantification in large language models (LLMs). Let $\mathcal{X}$ denote the space of all possible input prompts and $\mathcal{Y}$ the space of all possible text responses.

To assess the factuality of generated content, we adopt the concept of entailment (Mohri & Hashimoto, 2024). We formalize correctness constraints in terms of entailment with respect to some reference knowledge $y^*$. We define the *entailment operator* $\mathcal{E}\colon \mathcal{Y} \mapsto 2^{\mathcal{Y}}$ as:

$$\mathcal{E}(y) := \{y' \in \mathcal{Y} \colon y' \Rightarrow y\}, \tag{1}$$

where $y' \Rightarrow y$ indicates that $y'$ entails $y$, i.e., $\mathcal{E}(y)$ contains all statements that logically imply $y$.

We define a *split function* $S : \mathcal{Y} \to 2^{\mathcal{Y}}$ that decomposes a response into a set of atomic answers:

$$S(y) = \{c_1, c_2, \ldots, c_k\}, \tag{2}$$

where each $c_i$ is an individual factual answer made in $y$. Conversely, we define a *merge function* $M : 2^{\mathcal{Y}} \to \mathcal{Y}$ that combines a set of answers into a coherent response:

$$M(\{c_1, c_2, \ldots, c_k\}) = y, \tag{3}$$

where $y$ is a natural language text that integrates all answers $c_i$ in a coherent manner.

Given a ground truth reference $y^*$, a response $y$ is considered factually correct if and only if $y^* \in \mathcal{E}(M(S(y)))$, which is equivalent to $y^* \Rightarrow M(S(y))$. This reflects the notion that a response is factually correct if its component answers, when merged into a coherent statement, are entailed by the truth.

**Example:** Consider a ground truth $y^*$: "Paris is the capital of France. It has a population of approximately 2.2 million people and is home to the Eiffel Tower, which was completed in 1889." The response "Paris is the capital of France" is factually correct because $y^* \Rightarrow M(S(y))$, as this answer is directly supported by the ground truth. Similarly, "Paris is known for the Eiffel Tower, which was built in the 1880s" is also correct, as the completion year 1889 entails construction in the 1880s. However, the response "Paris is the capital of France and has a population of exactly 3 million people" is factually incorrect because $y^* \nRightarrow M(S(y))$, as the ground truth does not support the specific population answer.

Let $\{(X_i, y_i^*)\}_{i=1}^n$ represent our calibration dataset, where:

- $X_i \in \mathcal{X}$ denotes the input prompt
- $y_i^* \in \mathcal{Y}$ is the reference/ground truth answer to prompt $X_i$

Our goal is to develop a method that produces responses with a guaranteed level of factual correctness. Specifically, given a new input $X_{n+1}$, we aim to select a response such that the probability of it being factually correct is at least $1 - \alpha$ for a desired error rate $\alpha \in (0, 1)$.

### 3.2 CONFORMAL FACTUALITY

Our approach is based on split conformal prediction, which provides valid uncertainty quantification without distributional assumptions. In this setting, we split our data into a calibration set $\{(X_i, y_i^*)\}_{i=1}^n$ and a test set.

Given a nonconformity score function $r : \mathcal{X} \times \mathcal{Y} \to \mathbb{R}$ measuring the unusual nature of input-output pairs, the standard conformal prediction framework constructs a prediction set $\hat{C}_\alpha(X)$ for a new input $X$ such that:

$$P(y^* \in \hat{C}_\alpha(X)) \geq 1 - \alpha. \tag{4}$$

For a new test point $X_{n+1}$, we compute the conformal prediction set as:

$$\hat{C}_\alpha(X_{n+1}) = \{y \in \mathcal{Y} : r(X_{n+1}, y) \le \hat{q}_\alpha\}, \tag{5}$$

where $\hat{q}_\alpha$ is the $(1 - \alpha)$-quantile of the nonconformity scores on the calibration set $\{r(X_i, y_i^*)\}_{i=1}^n$.

In our LLM factuality setting, the connection to conformal prediction is direct: if $y_{n+1}$ is our calibrated model output for input $X_{n+1}$, then we want:

$$P(y_{n+1}^* \in \mathcal{E}(M(S(y_{n+1})))) \ge 1 - \alpha. \tag{6}$$

This guarantees that our model's calibrated output $y_{n+1}$, when processed through our split and merge functions, is factually correct with respect to the ground truth $y_{n+1}^*$ with probability at least $1 - \alpha$.

## 4 METHODOLOGY

### 4.1 RANK-BASED CONFORMAL PREDICTION FRAMEWORK

Our key innovation is a rank-based conformal prediction approach, **RankConf**, that leverages the LLM's ability to evaluate the quality of its own responses. Unlike existing approaches that use log-probability or perplexity, we define a novel nonconformity score based on the ranking of responses that captures the model's relative confidence in its generated responses. This method allows the calibrated LLM output to adapt naturally to input difficulty–providing precise answers for straightforward questions while appropriately hedging on challenging ones. Our ranking based approach is similar to the CDF-based conformity scores developed in (Dheur et al., 2025).

#### 4.1.1 COLLABORATIVE RESPONSE GENERATION AND RANKING PROCESS

For each input $X_i$ in our calibration set, our approach proceeds as follows:

1. Generate $K$ candidate responses using a **lower-version LLM** $L$: $\{L^{(1)}(X_i), L^{(2)}(X_i), \dots, L^{(K)}(X_i)\}$.

2. Construct an extended response set $R_i$ that includes both the generated responses and the ground truth answer:
$$R_i = \{L^{(1)}(X_i), L^{(2)}(X_i), \dots, L^{(K)}(X_i), y_i^*\}. \tag{7}$$

3. Have the **high-version LLM** ranks all responses in $R_i$ based on their perceived quality, assigning a rank : $\mathcal{Y} \to \{1, 2, \dots, K + 1\}$ to each response, where lower rank values indicate higher quality (rank 1 is best).

4. For each response $y \in R_i$, check whether the ground truth $y_i^*$ entails the response by evaluating whether $y_i^* \in \mathcal{E}(M(S(y)))$.

Our collaborative response generation and ranking design draws inspiration from speculative decoding strategies (Chen et al., 2023), creatively adapting this inference acceleration technique to uncertainty quantification. Rather than using small models to predict tokens for verification by larger models as in traditional speculative decoding, we employ lower-parameter LLMs to efficiently generate diverse candidate responses while leveraging higher-parameter models' superior evaluation capabilities to rank these responses based on factual accuracy. This approach aligns with cognitive science's dual-process theory, where rapid generation (system 1) is followed by analytical evaluation (system 2), and extends (Kadavath et al., 2022a) that language models possess inherent ability to assess their knowledge boundaries, but significantly enhances this capability through cross-model collaboration.

This design addresses the prohibitive cost of manual verification in specialized domains where human expertise is required, as expert validation of model outputs demands significant time and resources. It leverages the natural evolution of LLM platforms, where organizations typically maintain multiple model versions—newer, more capable models can serve as evaluators for previously deployed systems without requiring additional infrastructure. The dynamic threshold mechanism automatically adjusts response selection based on input difficulty, preserving more content for straightforward queries while applying stricter filtering for complex questions, thus optimizing the balance between information richness and factual reliability.

### 4.1.2 NON-CONFORMITY SCORE BASED ON RESPONSE RANKING

Given the ranked responses, we define our non-conformity score $r(X_i, y_i^*)$ as:

$$r(X_i, y_i^*) := \min\{\text{rank}(y) \mid y \in R_i, y_i^* \notin \mathcal{E}(M(S(y)))\} - 1. \tag{8}$$

This score represents the rank of the first factually incorrect response in the ranking, minus 1. Intuitively, it tells us how far down the ranked list we can go while still maintaining factual correctness. The subtraction of 1 accounts for the convention of returning the previous rank value when finding the first incorrect response.

We determine the conformal threshold based on the calibration set:

$$\hat{q}_\alpha = \text{Quantile}(\{r(X_i, y_i^*)\}_{i=1}^n, \lceil (n+1)(1-\alpha) \rceil / n). \tag{9}$$

Using this threshold, for a new input $X_{n+1}$, we define our calibrated prediction function $L^\alpha : \mathcal{X} \to \mathcal{Y}$ as:

$$L^\alpha(X_{n+1}) = M \left( \bigcup_{j:\text{rank}(L^{(j)}(X_{n+1})) \leq \hat{q}_\alpha} S(L^{(j)}(X_{n+1})) \right). \tag{10}$$

That is, $L^\alpha(X_{n+1})$ merges answers from all responses with ranks not exceeding our threshold $\hat{q}_\alpha$, creating a comprehensive answer that maintains the coverage guarantee.

**Theorem 4.1** (Factual Correctness Guarantee). *Let $\{(X_i, y_i^*)\}_{i=1}^{n+1}$ be exchangeable, and let $\hat{q}_\alpha$ be the $\lceil (n+1)(1-\alpha) \rceil / n$-quantile of $\{r(X_i, y_i^*)\}_{i=1}^n$. Then:*

$$P(y_{n+1}^* \in \mathcal{E}(M(S(L^\alpha(X_{n+1}))))) \geq 1 - \alpha. \tag{11}$$

*That is, the output of $L^\alpha(X_{n+1})$ is factually correct with probability at least $1 - \alpha$.*

*Proof.* Let $r_{n+1} = r(X_{n+1}, y_{n+1}^*)$. By the properties of conformal prediction and the exchangeability of the data, we have:

$$P(r_{n+1} \leq \hat{q}_\alpha) \geq 1 - \alpha. \tag{12}$$

By the definition of our non-conformity score, if $r_{n+1} \leq \hat{q}_\alpha$, then all responses $L^{(j)}(X_{n+1})$ with $\text{rank}(L^{(j)}(X_{n+1})) \leq \hat{q}_\alpha$ must be factually correct. For each such response, we have $y_{n+1}^* \in \mathcal{E}(M(S(L^{(j)}(X_{n+1}))))$. Since $L^\alpha(X_{n+1})$ merges answers from these factually correct responses, and the merge of factually correct answers maintains factual correctness, we have $y_{n+1}^* \in \mathcal{E}(M(S(L^\alpha(X_{n+1}))))$. Therefore:

$$P(y_{n+1}^* \in \mathcal{E}(M(S(L^\alpha(X_{n+1}))))) \geq P(r_{n+1} \leq \hat{q}_\alpha) \geq 1 - \alpha. \tag{13}$$

$\square$

### 4.2 COMPLETE ALGORITHM

Our algorithm, Algorithm 1 consists of two phases: *Calibration Phase* and *Prediction Phase*.

This algorithm, which we call **RankConf**, uses the language model's own ranking ability to determine which responses are likely to be factually correct, while providing a mathematical guarantee that the selected response is factually correct with probability at least $1 - \alpha$.

The empirical coverage of our method is defined as the fraction of $T$ test questions where the output is factually correct:

$$\text{Coverage} = \frac{1}{T} \sum_{t=1}^T \mathbb{1}\{y_{n+t}^* \in \mathcal{E}(M(S(L^\alpha(X_{n+t}))))\}, \tag{14}$$

where $T$ is the number of test questions and $\mathbb{1}\{\cdot\}$ is the indicator function.

---

**Algorithm 1 RankConf**: Rank-Based Conformal Factuality

---

1: **Input:** Calibration data $\{(X_i, y_i^*)\}_{i=1}^n$, language model $L$, confidence level $1 - \alpha$
2: **Output:** A prediction function $L^\alpha : \mathcal{X} \to \mathcal{Y}$ with factual correctness guarantee
3: **Calibration Phase:**
4: **for** $i = 1$ to $n$ **do**
5:     Low-version LLM generates $K$ responses $L^{(1)}(X_i), L^{(2)}(X_i), \ldots, L^{(K)}(X_i)$
6:     $R_i \leftarrow \{L^{(1)}(X_i), L^{(2)}(X_i), \ldots, L^{(K)}(X_i), y_i^*\}$
7:     High-version LLM obtains ranks for each response in $R_i$
8:     $r(X_i, y_i^*) \leftarrow \min\{\text{rank}(y) \mid y \in R_i, y_i^* \notin \mathcal{E}(M(S(y)))\} - 1$
9: $\hat{q}_\alpha \leftarrow \lceil(n+1)(1-\alpha)\rceil/n$-quantile of $\{r(X_i, y_i^*)\}_{i=1}^n$
10: **Prediction Phase:**
11: **for** new input $X_{n+1}$ **do**
12:     Low-version LLM generates $K+1$ responses $L^{(1)}(X_{n+1}), L^{(2)}(X_{n+1}), \ldots, L^{(K+1)}(X_{n+1})$
13:     High-version LLM obtains ranks for each response
14:     $L^\alpha(X_{n+1}) \leftarrow M\left(\bigcup_{j:\text{rank}(L^{(j)}(X_{n+1})) \leq \hat{q}_\alpha} S(L^{(j)}(X_{n+1}))\right)$
15:     **return** $L^\alpha(X_{n+1})$

---

### 4.3 OPTIMIZATION FRAMEWORK FOR ENHANCED CONDITIONAL COVERAGE

While our basic **RankConf** method provides marginal coverage guarantees, we can enhance conditional validity by adapting the threshold based on features of the input prompt. We call this enhanced method **AdaptiveRankConf**. Following (Cherian et al., 2024), we define an adaptive threshold function:

$$\hat{q}_\alpha(Z_{n+1}) = \sup\{r : r \leq g_r(Z_{n+1})\}, \tag{15}$$

where $Z_{n+1}$ are features computed from the input prompt $X_{n+1}$, and $g_r$ is obtained by solving:

$$g_r = \arg\min_{g \in \mathcal{F}} \frac{1}{n+1} \sum_{i=1}^n \ell_{\alpha(Z_i)}(r(X_i, y_i^*) - g(Z_i)) + \frac{1}{n+1} \ell_{\alpha(Z_{n+1})}(r - g(Z_{n+1})). \tag{16}$$

Here, $\mathcal{F}$ is a function class (e.g., linear functions of features), and $\ell_\alpha(\cdot)$ is the pinball loss at level $\alpha$.

The feature vector $Z_i$ is constructed through several components. First, we have the LLM categorize each of the $n$ questions into difficulty groups $G_i$ based on the question's topic, yielding grouping features $Z_i^G$. We also generate comprehensive answers for each calibration question and extract additional features including the question's main topic, average response length, average Wikipedia view count for related entities, and other metadata that may correlate with question difficulty. For specific feature selection details, please refer to the dataset introduction in Section 5.

This adaptive approach essentially employs a question difficulty estimator to produce an instance-specific threshold for the rank, allowing the threshold to vary based on the characteristics of each prompt. This provides stronger conditional validity guarantees across different domains and question types, as more difficult questions may require more conservative thresholds to maintain factual correctness. Furthermore, by making the error level $\alpha(Z)$ adaptive to input features, we can balance the trade-off between factuality and response quality dynamically.

## 5 EXPERIMENTAL SETUP

### 5.1 BASELINE METHODS AND OUR APPROACH

**SplitConf** (Mohri & Hashimoto, 2024): Employs sub-claim based filtering by decomposing responses into approximately 10 sub-claims per query and filtering them based on confidence scores derived from log-probability ratios of tokens. Implements static thresholds calibrated across the entire dataset.

**CondSplitConf** (Cherian et al., 2024): Also uses sub-claim based filtering but extends SplitConf with input-dependent thresholds using question topic metadata.

**RankConf (Ours)**: Our rank-based conformal framework (Algorithm 1) operates at the whole-response level, leveraging comparative judgment capabilities of LLMs to rank responses and establish factuality guarantees.

**AdaptiveRankConf (Ours)**: As described in Section 4.3, this enhancement incorporates input-dependent thresholds while maintaining our response-level approach.

## 5.2 DATASETS

**Datasets Split**  Unless otherwise specified, each experiment uses 50% for calibration and 50% for test, repeated over 50 random splits. For conditional validity assessment, we naturally utilize pre-grouped datasets such as MedicalQA, employing the difficulty levels already defined within the dataset. For ungrouped datasets, we defined groups based on question difficulty levels, such as (level 1, 2, 3) determined by LLM assessment of the complexity and specialized knowledge required.

**MedicalQA:** (Jeong et al., 2024) focuses on long-form medical question-answering tasks. It combines several established medical QA benchmark. This dataset comprises the following five categories: HEALTHSEARCH_QA, KQA_GOLDEN, KQA_SILVER, LIVE_QA, and MEDICATION_QA. We have naturally processed the dataset into five difficulty levels based on these categories as a feature vector.

**Natural Questions (NQ):** (Kwiatkowski et al., 2019) contains factual questions derived from Google search engine queries, designed for open-ended question answering evaluation. Since the dataset lacks a natural classification, we followed our previous design and divided it into three difficulty levels using LLM.

**FactScore:** (Min et al., 2023a) evaluates factual accuracy in open-ended generation by assessing claims against a comprehensive knowledge base. Following (Cherian et al., 2024), we grouped Wikipedia subjects by page view counts as a feature vector: "Very Frequent" ($\geq$1,000,000 views), "Frequent" (100,000-999,999 views), "Medium" (1,000-99,999 views), "Rare" (100-999 views), and "Very Rare" ($<$100 views).

**MATH:** (Hendrycks et al., 2021) comprises challenging mathematical problems that test reasoning capabilities, where answers involve sequential solution steps. The difficulty classification of MATH dataset is the same as NQ.

Following (Su et al., 2024), we generated experimental data using API Query interactions with real-world question datasets. Our process involved: (1) prompting models to categorize input questions by topic, difficulty level, and knowledge domain, with these categories serving as the groups for our conditional coverage analysis, (2) The lower-version model generates long-text responses and splits them into sub-answers. (3) The higher-version model provides high-quality rankings and entailment annotation. (4) Our Conformal process provides factual filtering, after which the model merges the filtered sub-answers to complete the output. The specific prompt design is given in Appendix B.

## 5.3 EVALUATION METRICS

**Marginal Coverage**: Percentage of test examples where the true response $y^*$ is included in the prediction set as in Equation (14).

**Coverage Gap (CovGap)**: Average absolute deviation between group-specific and target coverage across groups, measuring conditional validity:

$$\text{CovGap} = \frac{1}{|\mathcal{G}|} \sum_{g \in \mathcal{G}} |\text{Coverage}(g) - (1 - \alpha)|. \tag{17}$$

**Tail Coverage Rate (TCR)**: Mean coverage across hardest and easiest 10% of questions:

$$\text{TCR} = \frac{1}{2} \left( \frac{\sum_{s \in S_{\text{lower}}} \mathbb{1}\{y_s^* \in \hat{C}(X_s)\}}{0.1T} + \frac{\sum_{s \in S_{\text{upper}}} \mathbb{1}\{y_s^* \in \hat{C}(X_s)\}}{0.1T} \right), \tag{18}$$

where $S_{\text{lower}}$ and $S_{\text{upper}}$ contain the 10% of questions with highest and lowest nonconformity scores.

**Set Size and Retention Rate**: The size of the subanswers that can be returned to users after the test set samples have been factually verified.

$$\widehat{\mathcal{C}}_\alpha(X) = |L^{(j)}(X) \mid \text{rank}(L^{(j)}(X)) \leq \hat{q}_\alpha|. \tag{19}$$

A larger set size means that more subanswer are retained. Furthermore, we define the retention rate as follow,

$$\text{RetRate}(X_{n+1}) = \frac{|\hat{C}_\alpha(X_{n+1})|}{K}. \tag{20}$$

## 6 EXPERIMENTAL RESULTS

In this section, we present the following three main results: (i) marginal and conditional coverage metrics at different alpha levels, (ii) results for Set size and Retention rate, and (iii) ablation experiments examining the impact of different language model combinations on the $K$ (numbers of subanswers) and $T$ (model temperature) parameters.

Here, we use the MedicalQA and NQ dataset as a case study with **RankConf** corresponds to SplitConf, while **AdaptiveRCf** corresponds to CondSConf. Additional experimental results are reported in Appendix G. In all figures presenting results, shaded areas indicate the standard deviation of marginal coverage results in both positive and negative directions. In all tables, bolded data represents the optimal result, and underlined data indicates the second-best result. To simplify the description , we use I and II to represent the low-version and high-version models.

Our methods aim to (i) keep the same $(1 - \alpha)$ marginal coverage, (ii) improve conditional coverage, and (iii) under the user setting dynamic threshold, maximize the Set size and Retention rate.

Table 1: Experimental results of low-version model generation and high-version model ranking (I-Model=gemini-2.0-flash, II-Model=gemini-2.5-pro, $K = 50, T = 1, \alpha = 0.1$)

| Dataset | Method | Coverage | TCR@0.1 ↑ | CovGap↓ | Set size ↑ | RetRate(%) ↑ |
|---|---|---|---|---|---|---|
| **MedicalQA** | SplitConf | $0.911 \pm 0.027$ | 0.888 | 0.058 | $15.10 \pm 0.21$ | 30.24 |
| | **RankConf** | $0.909 \pm 0.023$ | 0.894 | 0.037 | $16.73 \pm 0.15$ | 33.48 |
| | CondSConf | $\underline{0.903 \pm 0.013}$ | **0.901** | $\underline{0.023}$ | $\underline{18.55 \pm 0.58}$ | $\underline{37.10}$ |
| | **AdaptiveRCf** | $\mathbf{0.901 \pm 0.008}$ | $\underline{0.902}$ | **0.011** | $\mathbf{19.28 \pm 0.70}$ | **38.56** |
| **NQ** | SplitConf | $0.912 \pm 0.015$ | 0.893 | 0.105 | $19.42 \pm 0.13$ | 38.84 |
| | **RankConf** | $0.907 \pm 0.021$ | **0.899** | 0.093 | $20.73 \pm 0.24$ | 41.46 |
| | CondSConf | $\underline{0.902 \pm 0.012}$ | 0.897 | $\underline{0.057}$ | $\underline{21.52 \pm 0.56}$ | $\underline{43.04}$ |
| | **AdaptiveRCf** | $\mathbf{0.900 \pm 0.009}$ | $\underline{0.901}$ | **0.051** | $\mathbf{23.11 \pm 0.10}$ | **46.22** |

**Core Indicators Performance.** As shown in Table 1, all methods achieve high and comparable coverage and TCR across both datasets, indicating that the overall ability to generate valid responses is well preserved regardless of the ranking strategy. Specifically, **RankConf**—our improvement over the baseline SplitConf—maintains similar coverage and TCR while enabling more informed selection. Likewise, our adaptive method **AdaptiveRCF** matches or slightly improves upon its counterpart CondSConf in these metrics. The differences become more pronounced in downstream effectiveness: **AdaptiveRCF** yields the largest average set size (e.g., 23.11 on NQ), suggesting it retains more diverse and potentially useful candidates, and consequently achieves the highest RetRate (46.22% on NQ and 38.56% on MedicalQA). In contrast, SplitConf and CondSConf produce smaller candidate sets and lower RetRate, discarding valuable outputs. In addition, we provide results on other datasets in Appendix B Table 3.

**Conditional Performance.** Figure 1 shows the marginal and conditional coverage of the four methods across question difficulty levels in MedicalQA and NQ. While all methods achieve marginal coverage close to the target $1 - \alpha$, their conditional coverage—especially for hard questions (Level 3)—differs markedly. SplitConf exhibits significant under-coverage on harder at high $\alpha$, whereas **RankConf** (our improvement) closes this gap by leveraging ranking information. Similarly, Cond-SConf improves over SplitConf but still falls short on difficult instances, while our **AdaptiveRCF** maintains near-ideal conditional coverage across all levels. In addition, we provide results on other datasets in Appendix B Table 2

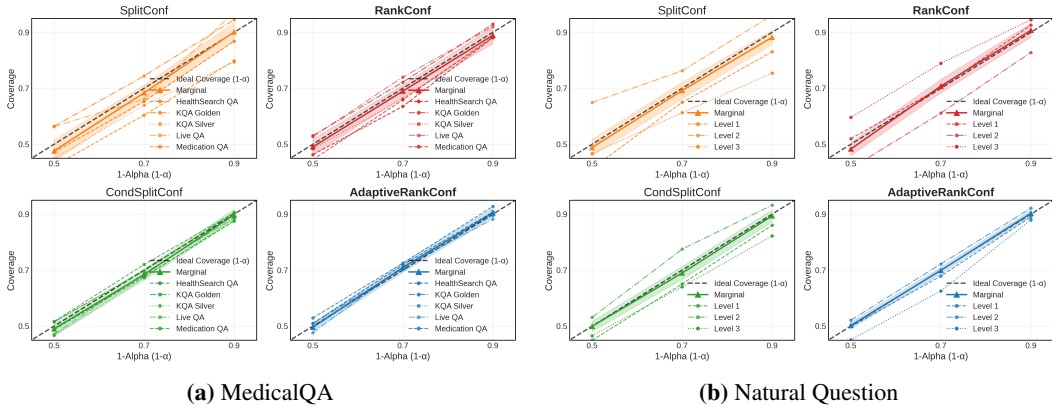

**(a)** MedicalQA  **(b)** Natural Question

Figure 1: Marginal and Conditional coverage of the four methods across three difficulty level groups in the MedicalQA and NQ dataset, for $\alpha$ values ranging from 0.5 to 0.9.

**Ablation Studies.** In addition to the default parameter settings used in the main experiments ($K = 50, T = 1$), we also evaluate performance across a broader range of configurations: $K \in \{10, 100\}$ and $T \in \{0.7, 1.5\}$. As shown in in Appendix B Table 3, **AdaptiveRCF** and **RankConf** consistently maintain high coverage and low CovGap across all these settings on both NQ and MedicalQA, demonstrating strong robustness to variations in candidate set size and generation temperature. This stability highlights that the adaptive and ranking-aware mechanisms in our methods effectively mitigate the impact of hyperparameter choices, making them more reliable in practical applications. Furthermore, we also evaluated the potential impact of different model combinations in Appendix B Table 4. The results demonstrate that our two proposed methods can produce optimal factual screening results even when applied to cross-platform model combinations.

## 7 CONCLUSION

In this paper, we propose a novel conformal prediction framework that quantifies uncertainty in language model text generation through collaborative ranking and dynamic thresholds. Our **RankConf** and **AdaptiveRankConf** employ ranking instead of relying on traditional probabilistic metrics. By having lower-tier LLMs generate candidate answers and higher-tier models rank them, we establish a robust factual filtering mechanism that adapts to varying input difficulty levels. This work provides a principled solution for deploying LLMs in high-stakes applications.

Limitations. Although our framework achieves significant progress in quantifying uncertainty in LLM contexts, several limitations warrant consideration.

1. Our approach assumes factual correctness can be reliably assessed through semantic entailment relationships, which may fail to capture all dimensions of truthfulness in complex reasoning tasks.

2. The method's effectiveness depends on the quality of the ranking model performance may decline when the gap between low- and high-ranking models is insufficient to capture subtle factual differences.

3. Computational overhead of generating and ranking multiple candidate answers may pose deployment challenges in latency-sensitive applications.

4. Our approach relies on the ranking during the LLM inference phase, which inherently limits its ability to address factual errors stemming from insufficient ranking during the LLM training phase or systemic biases.

Future work could explore more sophisticated difficulty estimation techniques and investigate extending our framework to multi-step reasoning scenarios where intermediate steps require separate quantification of uncertainty.

## REPRODUCIBILITY STATEMENT

Code is available at https://anonymous.4open.science/r/512499.The codebase includes implementations of our Algorithms, Model Query by API and Json dataset pre-processing code for our tasksand functions for computing the metrics and producing tables.

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

## A ADDITIONAL DETAILS FOR EXPERIMENT

**Default settings.** All experiments use listwise ranking prompts with explicit entailment rubrics; we standardize the merge prompt to enforce non-contradiction and de-duplication. Unless specified, $K=50$, $T=1.0$, $\alpha=0.1$, and top-$\hat{q}_\alpha(Z)$ selection uses features $Z=\{$question difficulty level, answer length, question type, log-prob$\}$.

**Diverse Model Combinations.** In our main experiments and ablation studies, we employed the following sets of LLM combinations. First, in Section 6 and the primary experiments, we used model pairs from the same platform: (1) I–Gemini-2.0-flash and II–Gemini-2.5-pro (Team et al., 2023), (2) I–Deepseek-V3 and II–Deepseek-R1 (Liu et al., 2024), (3) I–Qwen2.0-7B and II–Qwen3.0-7B (Yang et al., 2024). Additionally, in Appendix B Table 4, we supplemented our analysis with cross-platform model combinations to investigate potential data distribution shifts: (4) I–Gemini-2.0-flash and II–Qwen3.0-7B, (5) I–Qwen2.0-7B and II–Deepseek-R1, (6) I –Deepseek-V3 and II–Gemini-2.5-pro. This combinations design was guided by model release dates and parameter counts, under the general assumption that more recently released and larger-parameter models tend to exhibit stronger capabilities. We prompt each LLMs to generate a long-text response for each questions and decompose the original response into independent answers.

## B ADDITIONAL EXPERIMENT RESULTS

Table 2: Experimental results of low-version model generation and high-version model ranking (I-Model=gemini-2.0-flash, II-Model=gemini-2.5-pro, $K = 50, T = 1, \alpha = 0.1$)

| Dataset | Method | Coverage | TCR@0.1 ↑ | CovGap↓ | Set size ↑ | RetRate(%) ↑ |
|---|---|---|---|---|---|---|
| **Fastscore** | SplitConf | $0.912 \pm 0.022$ | 0.923 | 0.105 | $25.10 \pm 0.23$ | 50.20 |
| | **RankConf** | $0.907 \pm 0.017$ | 0.905 | 0.093 | $26.39 \pm 0.15$ | 53.46 |
| | CondSConf | $\underline{0.902 \pm 0.019}$ | **0.908** | $\underline{0.051}$ | $\underline{27.22 \pm 0.50}$ | $\underline{57.08}$ |
| | **AdaptiveRCf** | $\mathbf{0.899 \pm 0.013}$ | $\underline{0.902}$ | **0.051** | $\mathbf{29.30 \pm 0.70}$ | **57.65** |
| **MATH** | SplitConf | $0.883 \pm 0.037$ | 0.880 | 0.93 | $10.21 \pm 0.020$ | 20.42 |
| | **RankConf** | $0.905 \pm 0.020$ | **0.905** | 0.059 | $12.73 \pm 0.47$ | 25.46 |
| | CondSConf | $\underline{0.898 \pm 0.013}$ | 0.901 | $\underline{0.047}$ | $\underline{12.10 \pm 0.21}$ | $\underline{24.20}$ |
| | **AdaptiveRCf** | $\mathbf{0.897 \pm 0.007}$ | $\underline{0.896}$ | **0.031** | $14.73 \pm 0.15$ | **29.46** |

### B.1 ABLATION ACROSS DIVERSE MODEL PAIRS

To evaluate the robustness of our methods under varying model capabilities and potential distribution shifts, we conducted ablation studies across diverse model pairings. Table 4 presents results across six different model pairs using the MedicalQA dataset. Across all settings, **AdaptiveRCF** consistently achieves the lowest CovGap while maintaining the highest Set Size and Retention Rate, demonstrating remarkable robustness to differences in model capability and architecture. Notably, even when the II-Model is from a different architecture or training paradigm (cross-platform combinations), **AdaptiveRCF** preserves its superior conditional reliability, as evidenced by near-optimal Coverage and TCR values that consistently rank first or second in proximity to the ideal 0.9 target. Similarly, **RankConf** consistently outperforms the baseline SplitConf across both Coverage and TCR, validating the fundamental benefit of incorporating ranking signals into the confidence calibration process. These results collectively confirm that adaptive thresholding and ranking-aware strategies are essential components for effective uncertainty quantification, particularly in practical deployment scenarios involving heterogeneous or black-box LLM systems.

### B.2 FEATURE ABLATION AND COEFFICIENT ANALYSIS

To systematically evaluate feature contributions to **AdaptiveRankConf**'s coverage guarantee, we conducted comprehensive ablation studies and coefficient analysis across multiple datasets. As shown in Table 5, our method achieves the smallest coverage gap (CovGap) when using the complete feature set, confirming the synergistic value of our feature combination strategy.

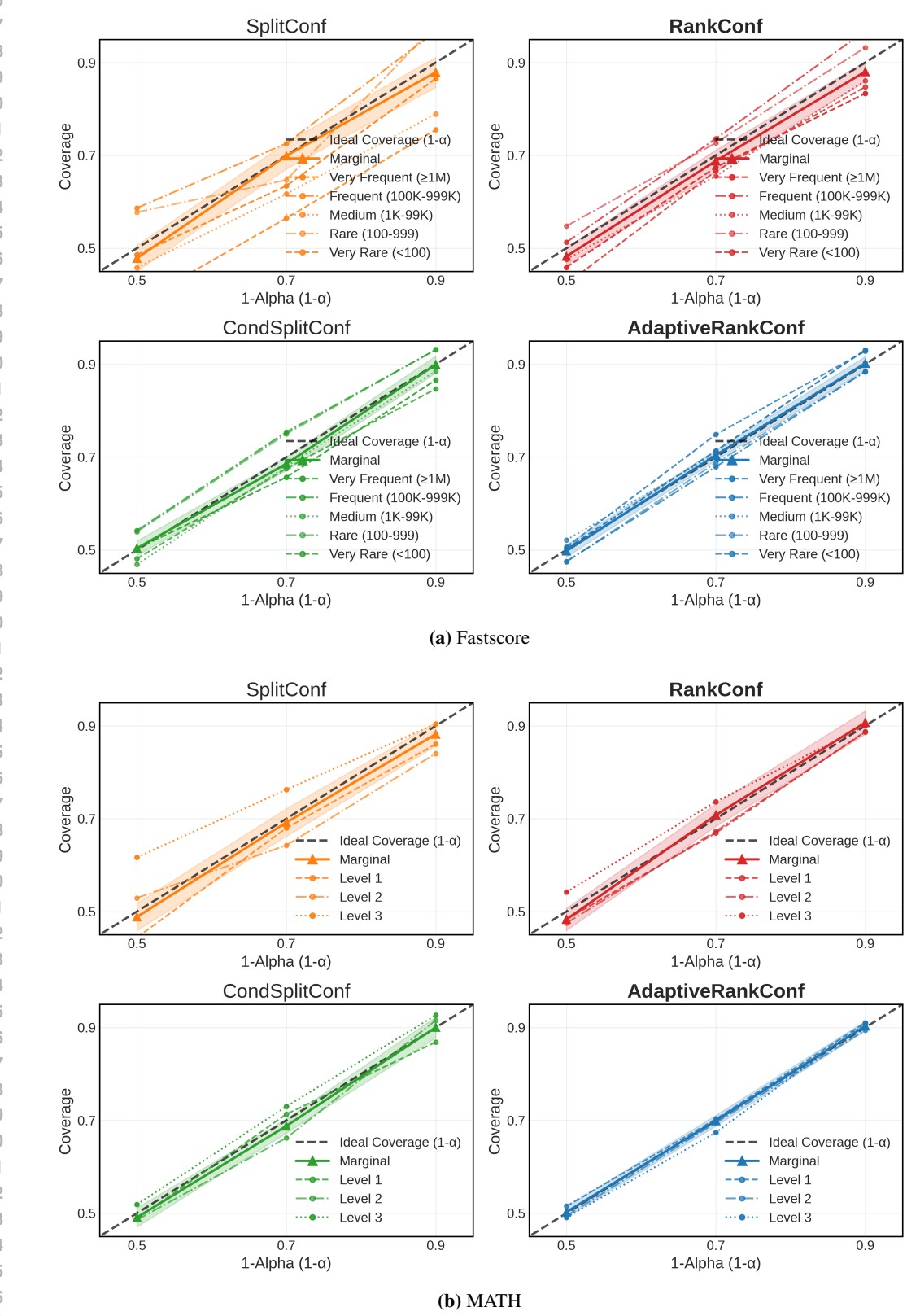

(a) Fastscore

(b) MATH

Figure 2: Marginal and Conditional coverage of the four methods in the Fastscore and NQ dataset, for $\alpha$ values ranging from 0.5 to 0.9.

Table 3: Ablation Experimental Results on NQ and MedicalQA with Different $K$ and $T$ Settings (I-Model=gemini-2.0-flash, II-Model=gemini-2.5-pro, $\alpha = 0.1$).

| Dataset | $K$ | $T$ | Metric | SplitConf | RankConf | CondSConf | AdaptiveRCF |
|---|---|---|---|---|---|---|---|
| MedicalQA | 10 | 1.0 | Coverage
Set size↑
CovGap↓ | $0.885 \pm 0.210$
$4.23 \pm 0.05$
0.052 | $0.908 \pm 0.050$
$4.67 \pm 0.02$
0.045 | $0.912 \pm 0.042$
$6.07 \pm 0.72$
0.026 | $\mathbf{0.902 \pm 0.210}$
$\mathbf{6.72 \pm 0.42}$
**0.013** |
| | 100 | 1.0 | Coverage
Set size↑
CovGap↓ | $0.906 \pm 0.026$
$18.27 \pm 0.17$
0.054 | $0.891 \pm 0.035$
$18.91 \pm 0.12$
0.039 | $0.899 \pm 0.027$
$26.74 \pm 0.24$
0.031 | $\mathbf{0.900 \pm 0.014}$
$\mathbf{28.12 \pm 0.41}$
**0.029** |
| | 50 | 0.7 | Coverage
Set size↑
CovGap↓ | $0.897 \pm 0.040$
$15.08 \pm 0.05$
0.039 | $0.902 \pm 0.003$
$15.12 \pm 0.65$
0.035 | $0.896 \pm 0.014$
$19.32 \pm 0.62$
0.012 | $\mathbf{0.901 \pm 0.090}$
$\mathbf{19.58 \pm 0.25}$
**0.011** |
| | 50 | 1.5 | Coverage
Set size↑
CovGap↓ | $0.894 \pm 0.630$
$13.91 \pm 0.20$
0.076 | $0.902 \pm 0.310$
$13.11 \pm 0.13$
0.052 | $\mathbf{0.900 \pm 0.217}$
$15.51 \pm 0.33$
**0.035** | $0.901 \pm 0.180$
$\mathbf{16.18 \pm 0.70}$
0.035 |
| NQ | 10 | 1.0 | Coverage
Set size↑
CovGap↓ | $0.922 \pm 0.007$
$5.27 \pm 0.17$
0.079 | $0.903 \pm 0.012$
$5.97 \pm 0.06$
0.053 | $0.918 \pm 0.022$
$6.07 \pm 0.24$
0.015 | $\mathbf{0.902 \pm 0.008}$
$\mathbf{6.52 \pm 0.08}$
**0.014** |
| | 100 | 1.0 | Coverage
Set size↑
CovGap↓ | $0.904 \pm 0.044$
$25.20 \pm 0.24$
0.031 | $0.900 \pm 0.510$
$26.91 \pm 0.34$
0.029 | $0.902 \pm 0.620$
$32.69 \pm 0.72$
0.011 | $\mathbf{0.900 \pm 0.260}$
$\mathbf{37.42 \pm 0.25}$
**0.009** |
| | 50 | 0.7 | Coverage
Set size↑
CovGap↓ | $0.906 \pm 0.102$
$16.53 \pm 0.75$
0.073 | $0.898 \pm 0.052$
$17.33 \pm 0.61$
0.051 | $0.903 \pm 0.084$
$19.32 \pm 0.33$
0.015 | $\mathbf{0.900 \pm 0.053}$
$\mathbf{22.36 \pm 0.87}$
**0.011** |
| | 50 | 1.5 | Coverage
Set size↑
CovGap↓ | $0.889 \pm 0.021$
$14.71 \pm 0.37$
0.086 | $0.895 \pm 0.101$
$14.90 \pm 0.51$
0.062 | $0.895 \pm 0.042$
$17.30 \pm 0.32$
0.030 | $\mathbf{0.902 \pm 0.016}$
$\mathbf{18.61 \pm 0.47}$
**0.028** |

Table 4: Experimental results using the MedicalQA dataset for different LLM pair combinations: (1),(2),(3), represent combinations of models from the same platform, while (4),(5),(6) represent combinations of models from different platforms. ($K = 50, T = 1, \alpha = 0.1$).

| Model Pairs | Method | Coverage | TCR@0.1 | CovGap↓ | Set Size ↑ | RetRate(%) ↑ |
|---|---|---|---|---|---|---|
| (1) I–**Gemini-2.0-flash** and II–**Gemini-2.5-pro** | SplitConf
RankConf
CondSConf
AdaptiveRCf | $0.911 \pm 0.027$
$0.909 \pm 0.023$
$0.903 \pm 0.020$
$\mathbf{0.901 \pm 0.012}$ | 0.888
0.894
0.901
**0.902** | 0.058
0.037
0.023
**0.011** | $15.10 \pm 0.21$
$16.73 \pm 0.15$
$18.55 \pm 0.58$
$\mathbf{19.28 \pm 0.70}$ | 30.24
33.48
37.10
**38.56** |
| (2)I–**Deepseek-V3** and II–**Deepseek-R1** | SplitConf
RankConf
CondSConf
AdaptiveRCf | $0.908 \pm 0.023$
$0.905 \pm 0.018$
$0.902 \pm 0.020$
$\mathbf{0.900 \pm 0.005}$ | 0.912
0.903
0.899
**0.901** | 0.065
0.044
0.031
**0.020** | $15.64 \pm 0.47$
$17.08 \pm 0.07$
$19.31 \pm 0.20$
$\mathbf{20.03 \pm 0.25}$ | 31.28
34.16
38.62
**40.06** |
| (3)I–**Qwen2.0-7B** and II–**Qwen3.0-7B** | SplitConf
RankConf
CondSConf
AdaptiveRCf | $0.915 \pm 0.053$
$0.912 \pm 0.018$
$0.904 \pm 0.012$
$\mathbf{0.905 \pm 0.009}$ | 0.920
0.913
0.903
**0.903** | 0.138
0.088
0.064
**0.047** | $14.40 \pm 0.93$
$14.89 \pm 0.55$
$\mathbf{19.01 \pm 0.12}$
$18.92 \pm 0.34$ | 28.80
29.78
**38.02**
37.84 |
| (4)I–**Gemini-2.0-flash** and II–**Qwen3.0-7B** | SplitConf
RankConf
CondSConf
AdaptiveRCf | $0.885 \pm 0.025$
$0.898 \pm 0.022$
$0.901 \pm 0.013$
$\mathbf{0.901 \pm 0.010}$ | 0.878
0.896
0.902
**0.901** | 0.189
0.133
0.083
**0.064** | $13.50 \pm 0.30$
$15.23 \pm 0.34$
$\mathbf{17.13 \pm 0.32}$
$17.02 \pm 0.53$ | 27.15
30.46
**34.26**
34.04 |
| (5)I–**Qwen2.0-7B** and II–**Deepseek-R1** | SplitConf
RankConf
CondSConf
AdaptiveRCf | $0.882 \pm 0.033$
$0.893 \pm 0.028$
$0.895 \pm 0.029$
$\mathbf{0.898 \pm 0.022}$ | 0.888
0.895
0.897
**0.898** | 0.237
0.151
0.083
**0.066** | $13.46 \pm 0.77$
$14.83 \pm 0.43$
$\mathbf{16.73 \pm 0.68}$
$16.92 \pm 0.20$ | 26.92
29.66
**33.46**
33.84 |
| (6)I–**Deepseek-V3** and II–**Gemini-2.5-pro** | SplitConf
RankConf
CondSConf
AdaptiveRCf | $0.882 \pm 0.023$
$0.893 \pm 0.012$
$\mathbf{0.894 \pm 0.022}$
$0.895 \pm 0.009$ | 0.890
0.894
0.893
**0.895** | 0.209
0.142
0.069
**0.060** | $15.48 \pm 0.53$
$16.19 \pm 0.31$
$\mathbf{18.51 \pm 0.64}$
$18.83 \pm 0.81$ | 30.96
32.38
**37.02**
37.66 |

Table 5: Ablation study of feature contributions to AdaptiveRCf performance

| Datasets | Metric | I – Full Feature Set | II – w/o Question difficulty | III – w/o Question type | IV – w/o LLM log-prob | V – w/o Answer length |
|---|---|---|---|---|---|---|
| MedicalQA | Coverage | 0.901 ± 0.008 | 0.897 ± 0.02 | 0.905 ± 0.61 | 0.899 ± 0.24 | 0.904 ± 0.35 |
| | Set Size ↑ | 19.28 ± 0.70 | 16.43 ± 0.42 | 18.53 ± 0.09 | 16.95 ± 0.26 | 18.84 ± 0.20 |
| | CovGap ↓ | 0.011 | 0.031 | 0.019 | 0.026 | 0.023 |

| Datasets | Metric | I – Full Feature Set | II – w/o Wikipedia view counts | III – w/o Question type | IV – w/o LLM log-prob | V – w/o Answer length |
|---|---|---|---|---|---|---|
| Factscore | Coverage | 0.899 ± 0.013 | 0.895 ± 0.13 | 0.903 ± 0.61 | 0.898 ± 0.46 | 0.900 ± 0.83 |
| | Set Size ↑ | 29.30 ± 0.70 | 9.68 ± 0.24 | 12.52 ± 0.24 | 10.02 ± 0.57 | 11.68 ± 0.31 |
| | CovGap ↓ | 0.051 | 0.084 | 0.065 | 0.079 | 0.081 |

Table 6: Coefficient analysis of the experiment using features

| Dataset | Feature | Coefficient ± SE | Standardized Beta | t-value | p-value |
|---|---|---|---|---|---|
| MedicalQA | Intercept | -0.79 ± 0.05 | – | -12.15 | < 0.001 |
| | Question difficulty level | **0.39 ± 0.04** | **0.28** | **8.13** | < 0.001 |
| | Question type | 0.09 ± 0.06 | 0.07 | 1.63 | 0.105 |
| | LLM log-prob | **0.37 ± 0.04** | **0.32** | **9.74** | < 0.001 |
| | Answer length | 0.08 ± 0.02 | 0.15 | 2.86 | 0.005 |
| Factscore | Intercept | -0.83 ± 0.07 | – | -11.86 | < 0.001 |
| | Wikipedia view count | **0.42 ± 0.05** | **0.31** | **8.4** | < 0.001 |
| | Question type | 0.11 ± 0.06 | 0.08 | 1.81 | 0.072 |
| | LLM log-prob | **0.35 ± 0.04** | **0.29** | **8.75** | < 0.001 |
| | Answer length | 0.09 ± 0.03 | 0.17 | 2.98 | 0.004 |

To quantify feature importance, we performed multivariate regression analysis establishing the relationship between feature vectors $Z_i$ and nonconformity scores $r(X_i, y_i^*)$:

$$r(X_i, y_i^*) = \beta_0 + \sum_{j=1}^{p} \beta_j Z_{i,j} + \epsilon_i, \quad \beta_j^{\text{std}} = \beta_j \cdot \frac{\sigma_{Z_j}}{\sigma_r}, \quad t_j = \frac{\beta_j}{\text{SE}(\beta_j)} \tag{21}$$

Analysis of Table 6 reveals consistent feature importance patterns across domains. Crucially, removing Wikipedia page views (a key external knowledge feature) or LLM self-reported difficulty (a critical internal uncertainty metric) leads to substantial CovGap increases (0.033 and 0.014 on Factscore), demonstrating these features' essential role in conditional coverage. The coefficient analysis confirms both features exhibit strong positive predictive weights ($p < 0.001$) across datasets.

Question difficulty level significantly impacts threshold determination in domain-specific datasets (MedicalQA), while answer length shows moderate but consistent significance across all settings. In contrast, question type exhibits weaker predictive power ($p > 0.05$ in most cases), indicating its secondary importance in threshold adaptation.

These complementary analyses validate our dual-source approach that integrates external knowledge accessibility with internal model uncertainty. This integration enables dynamic threshold adjustment that responds to input characteristics, explaining our method's superior conditional coverage performance across diverse question types and domains. Statistical significance ($p < 0.001$) for core features provides strong evidence for the importance of feature-informed threshold adaptation in conformal language generation.

## C  PROMPTS AND QUERY DESIGN PROCESS

Table 7: Collaborative Model Prompt Design: Prompts for Subclaim Generation and Annotation

---

**Stage 1: Low-version Model - Subclaim Generation (gemini-2.0-flash-exp)**

**System Prompt:** "You are a highly intelligent medical AI assistant. Your task is to provide comprehensive medical information and break it down into structured subclaims."

**Task 1: Detailed Response Generation**
Provide a detailed English medical response for the medical question. Cover all relevant medical aspects, treatments, symptoms, causes, and recommendations. Ensure the answer is medically accurate and well-structured.

**Task 2: Subclaim Decomposition**
Decompose your comprehensive answer into $K$ distinct subclaims. Each subclaim should be a complete, standalone medical statement. Subclaims should follow the logical flow of your comprehensive answer. Each subclaim should be 10-30 words long for clarity.

**Output Format:** JSON object containing: Question, Free_form_answer, Must_have, Nice_to_have, Overall_length, claims[subclaim_seq{N}, Related_context], difficulty, source

---

**Stage 2: High-version Model - Annotation and Ranking (gemini-2.5-flash-lite-preview)**

**System Prompt:** "You are an expert medical evaluator. Your task is to analyze and evaluate medical subclaims for accuracy, relevance, and quality."

**Task 1: Subclaim Annotation Based on Reference Answer**
Evaluate the correctness of each subclaim using the reference answer below. Mark each subclaim as "True" (T) if it contains semantic content aligning with any of the "Must_have" or "Nice_to_have" lists, or "False" (F) if it doesn't contain such information. Add annotation to each subclaim in the JSON, with values "T" or "F".
*Reference Information:*
*Must_have:* Information that is essential and must be included
*Nice_to_have:* Information that is beneficial but not mandatory

**Task 2: Subclaim Ranking Based on Quality and Completeness**
Rank the input subclaims for quality and information completeness using the following criteria:
(1) Subclaims with high generation quality and good information completeness should rank higher.
(2) Rankings must be unique positive integers from 1 to the total number of subclaims, where 1 is the best rank, and higher numbers indicate lower quality.
(3) Subclaims with annotation "T" should generally rank significantly higher than those with "F", though not absolutely.
Add a 'rank' field to each subclaim in the JSON, with the rank value as a string.

**Task 3: Reconstruct Answer from High-Ranking Subclaims**
Identify all subclaims with ranks above a specified threshold. Semantically merge these subclaims to form the final correct answer. Return the reconstructed answer as a single continuous text, without adding extra content.

Table 8: Illustrative example of the collaborative ranking process on MedicalQA dataset. The table demonstrates how our **AdaptiveRankConf** method identifies factual Subclaims through this process desgin, filters out incorrect information, and reconstructs a coverage-guaranteed answer.

| Component | Content |
|---|---|
| **Question** | What is prednisone used for? |
| **Entailment Reference** | 1.Prednisone is a type of corticosteroid medication that is used to suppress the immune system and decrease inflammation in the body. 2.It's used to treat a host of different diseases and conditions, largely those associated with inflammation and immune system activity. 3.Common uses for prednisone include treating autoimmune diseases. Prednisone can also be used to prevent the body from rejecting an organ after a transplant. |
| **Low-version LLM Response** | Prednisone is a corticosteroid medication used primarily to suppress the immune system and decrease inflammation throughout the body. These properties make it useful in treating various conditions, including autoimmune diseases, severe allergies, and asthma. Although it does not specifically suppress the immune response to the new organ, it can also be used to manage certain types of cancers and may be prescribed to prevent organ rejection after transplant, It is not typically part of a chemotherapy regimen for specific cancers, nor are its immunosuppressant effects considered key in transplant settings, and it is not commonly used for certain eye conditions. |
| **High-version LLM Ranked Subclaims** | [T] 1. Prednisone is a corticosteroid medication used primarily to suppress the immune system. 
 [T] 2. It also serves to decrease inflammation throughout the body. 
 [T] These properties make it useful in treating various conditions, including autoimmune diseases. 
 [T] 3. Prednisone is also prescribed for severe allergies and asthma. 
 [T] 4. It can also be used to manage certain types of cancers. 
 [T] 5. Prednisone may be used to prevent organ rejection after transplant. |
| **AdaptiveRankConf Dynamic Threshold Calibration** | {"dataset": "MedicalQA", "difficulty_level": "Level 3", "dynamic_threshold": "rank_6"}The dynamic threshold is derived from the calibration set via $\hat{q}(Z)$, ensuring adaptive selection based on question difficulty and model confidence. |
| **Excluded Subclaims** | [F] 6. It suppresses the immune response to the new organ. 
 [F] 7. It may be part of a chemotherapy regimen for specific cancers. 
 [F] 8. Prednisone's immunosuppressant effects are key in transplant settings. 
 [F] 9. It's also sometimes used for certain eye conditions. |
| **Re-merge Answer** | Prednisone is a corticosteroid medication used primarily to suppress the immune system and decrease inflammation throughout the body. These properties make it useful in treating various conditions, including autoimmune diseases, severe allergies, and asthma. ~~Although it does not specifically suppress the immune response to the new organ,~~ It can also be used to manage certain types of cancers and may be prescribed to prevent organ rejection after transplant. ~~It is not typically part of a chemotherapy regimen for specific cancers, nor are its immunosuppressant effects considered key in transplant settings, and it is not commonly used for certain eye conditions.~~ |

# D  LARGE MODEL COLLABORATION AND RESOURCE ANALYSIS

In this section, we present a detailed empirical analysis of computational resource consumption across different model collaboration strategies. Our evaluation focuses on token usage and wall-clock time per query, measured under consistent experimental conditions. All computational resource metrics (token consumption and processing time) are reported as averages over 50 random train–calibration–test splits, with variability captured by standard deviation. Specifically:

1. **Token counting**: We obtain precise token statistics via the usage field returned by each API call. For a given input query, we first issue a single batch request (batch size = 1) to the low-version LLM to generate $K$ candidate responses, and record the total input and output tokens. We then issue a single batch request to the high-version LLM for ranking and entailment-based annotation, again recording total tokens. The sum from both calls constitutes the total token cost per query. Tokenization is performed using each model provider's native tokenizer to ensure cross-model consistency.

2. **Time measurement protocol**: Processing time is measured from the moment the API request is sent to the instant the full response is received, inclusive of network transmission latency.

**Remark.** Comprehensive token cost comparisons heavily depend on specific deployment environments and API providers. In this experiment, we used the default API settings of each model provider for testing. The cost analysis here is for reference only, and the actual cost may vary depending on the code runtime environment.

Table 9: Resource efficiency and performance across configurations. $L$-$H$ = Low-gen + High-rank, $L$-$L$ = Low-gen + Low-rank, $H$-$H$ = High-gen + High-rank. Set size denotes the number of factually verified subclaims retained. All results averaged over 50 random splits. (I-Model=gemini-2.0-flash, II-Model=gemini-2.5-pro, $\alpha = 0.1$)

| Dataset | $K$ | Config | Coverage | Set size ↑ | CovGap ↓ | Tokens (total) | Time (s/query) |
|---------|-----|--------|----------|-----------|----------|----------------|----------------|
| **MedicalQA** | 10 | *L-H* | **0.908±0.050** | 4.67±0.02 | **0.045** | 3,627 | 3.48 |
| | | *L-L* | 0.872±0.065 | 3.75±0.05 | 0.085 | **1,423** | **1.32** |
| | | *H-H* | 0.915±0.038 | 4.21±0.03 | 0.058 | 5,847 | 4.65 |
| | 50 | L-H | **0.909±0.023** | **16.73±0.15** | 0.040 | 12,483 | 6.78 |
| | | *L-L* | 0.865±0.043 | 12.40±0.20 | 0.092 | 5,732 | 2.53 |
| | | *H-H* | 0.921±0.027 | 15.42±0.21 | 0.055 | 34,215 | 15.87 |
| | 100 | *L-H* | **0.891±0.035** | **18.91±0.12** | 0.042 | 20,674 | 13.42 |
| | | *L-L* | 0.850±0.040 | 14.60±0.25 | 0.105 | 10,587 | 4.86 |
| | | *H-H* | 0.924±0.019 | 17.86±0.18 | 0.058 | 67,389 | 17.75 |
| **FactScore** | 10 | *L-H* | 0.907±0.017 | 26.39±0.15 | 0.093 | 2,547 | 2.38 |
| | | *L-L* | 0.875±0.032 | 22.15±0.22 | 0.125 | **973** | **0.87** |
| | | *H-H* | **0.913±0.025** | 24.53±0.18 | 0.102 | 4,038 | 3.15 |
| | 50 | *L-H* | 0.907±0.017 | **26.39±0.15** | 0.093 | 8,463 | 4.58 |
| | | *L-L* | 0.868±0.029 | 21.48±0.31 | 0.142 | 3,967 | 1.68 |
| | | *H-H* | **0.915±0.021** | 25.17±0.24 | 0.108 | 23,245 | 10.68 |
| | 100 | *L-H* | **0.900±0.510** | **26.91±0.34** | **0.029** | 13,872 | 8.93 |
| | | *L-L* | 0.857±0.035 | 21.93±0.28 | 0.158 | 7,263 | 3.28 |
| | | *H-H* | 0.918±0.018 | 26.03±0.32 | 0.113 | 46,128 | 17.27 |

Experimental results demonstrate that the *L-L* configuration (low-capability generation with low-capability ranking) achieves the lowest computational cost but suffers from insufficient coverage (0.872) and substantial conditional coverage gaps (CovGap 0.085), rendering it inadequate for high-stakes domains requiring rigorous factual accuracy. In contrast, the *H-H* configuration (high-capability generation with high-capability ranking) delivers strong coverage but incurs excessive computational overhead, resulting in significant resource inefficiency.

The *L-H* configuration (low-capability generation with high-capability ranking) achieves an optimal balance between resource efficiency and factual reliability: it consumes approximately 50% fewer tokens than *H-H* while maintaining near-optimal coverage and yielding the highest retention rate of

factually verified content. These findings validate our method efficiency—that offloading candidate generation to lightweight models while reserving critical factual verification and ranking tasks for more capable models represents the most effective strategy for building efficient and reliable conformal language generation systems.

# E    DETAILED PROOF OF THEOREM 4.1

*Proof.* To establish the factual correctness guarantee for our conformal prediction framework, we must rigorously demonstrate that the output of $L_\alpha(X_{n+1})$ is factually correct with probability at least $1 - \alpha$.

Let $r(X_i, y_i^*)$ denote the nonconformity score for the $i$-th calibration example, defined as:

$$r(X_i, y_i^*) := \min\{\text{rank}(y) \mid y \in R_i, y_i^* \notin \mathcal{E}(M(S(y)))\} - 1$$

Let $q_\alpha$ be the $\lceil (n+1)(1-\alpha) \rceil / n$-quantile of the nonconformity scores $\{r(X_i, y_i^*)\}_{i=1}^n$, and let $\hat{q}_\alpha$ be its empirical estimate computed from the calibration set.

Now, consider a new test example $(X_{n+1}, y_{n+1}^*)$, where $r_{n+1} = r(X_{n+1}, y_{n+1}^*)$. The fundamental property of conformal prediction guarantees that:

$$\mathbb{P}(r_{n+1} \leq \hat{q}_\alpha) \geq 1 - \alpha$$

To prove the claim in Theorem 4.1, it suffices to show that:

$$r_{n+1} \leq \hat{q}_\alpha \iff y_{n+1}^* \in \mathcal{E}(M(S(L_\alpha(X_{n+1}))))$$

For the forward direction, assume $r_{n+1} \leq \hat{q}_\alpha$. By definition of the nonconformity score, this means the rank of the first factually incorrect response in the ordered set $R_{n+1}$ is at least $\hat{q}_\alpha + 1$. Therefore, all responses $L^{(j)}(X_{n+1})$ with $\text{rank}(L^{(j)}(X_{n+1})) \leq \hat{q}_\alpha$ must be factually correct (i.e., $y_{n+1}^* \in \mathcal{E}(M(S(L^{(j)}(X_{n+1}))))$ for all such $j$).

Since $L_\alpha(X_{n+1})$ is defined as:

$$L_\alpha(X_{n+1}) = M \left( \bigcup_{j: \text{rank}(L^{(j)}(X_{n+1})) \leq \hat{q}_\alpha} S(L^{(j)}(X_{n+1})) \right)$$

the merged response contains only factually correct subclaims. By the properties of the entailment operator $\mathcal{E}$ and merge function $M$, the combined response must also be factually correct. Therefore, $y_{n+1}^* \in \mathcal{E}(M(S(L_\alpha(X_{n+1}))))$.

For the reverse direction, we prove the contrapositive. Assume $y_{n+1}^* \notin \mathcal{E}(M(S(L_\alpha(X_{n+1}))))$. This means the merged response contains at least one factually incorrect subclaim. Let $j^*$ be the index of the first factually incorrect response in the ranking of $R_{n+1}$. Then by definition of the nonconformity score:

$$r_{n+1} = \text{rank}(L^{(j^*)}(X_{n+1})) - 1$$

Since $y_{n+1}^* \notin \mathcal{E}(M(S(L_\alpha(X_{n+1}))))$, we must have $\text{rank}(L^{(j^*)}(X_{n+1})) \leq \hat{q}_\alpha + 1$, which implies $r_{n+1} \leq \hat{q}_\alpha$.

Combining these directions, we have established that:

$$r_{n+1} \leq \hat{q}_\alpha \iff y_{n+1}^* \in \mathcal{E}(M(S(L_\alpha(X_{n+1}))))$$

Therefore, by the properties of conformal prediction:

$$\mathbb{P}(y_{n+1}^* \in \mathcal{E}(M(S(L_\alpha(X_{n+1}))))) = \mathbb{P}(r_{n+1} \leq \hat{q}_\alpha) \geq 1 - \alpha$$

This completes the proof that our method provides the desired factual correctness guarantee.    □

The proof leverages the fundamental properties of conformal prediction while carefully accounting for our novel ranking-based nonconformity score and the specific definitions of our entailment-based factual correctness criterion.

# F    EXTENSION TO MULTI-STEP REASONING TASKS

Following the methods we described earlier, we will extend the collaborative ranking and dynamic thresholds approach to deep thinking models and tasks that require multi-step reasoning. For multi-step reasoning tasks, we consider problems that require $H$ sequential inference steps. At each step $h$, the model generates intermediate claims based on the initial prompt and previous reasoning steps. Formally, we denote:

- $X$: The initial input prompt
- $S_h$: The set of intermediate subclaims generated at step $h$
- $y_h^*$: The ground truth reference for step $h$
- $S_{1:h-1}$: The concatenation of all previous reasoning steps

The complete reasoning trajectory is represented as $S = (S_1, S_2, \ldots, S_H)$. For calibration example $i$, we construct the extended response set:

$$R_i = \left\{ (S_1^{(j_1)}, \ldots, S_H^{(j_H)}) \mid 1 \leq j_h \leq K, \forall h \in \{1, \ldots, H\} \right\} \cup \{(y_{i,1}^*, \ldots, y_{i,H}^*)\}$$

where $K$ is the number of candidate subclaim sets generated by the low-version LLM at each step, and the high-version LLM evaluates the entire trajectory.

## F.1    COVERAGE GUARANTEES FOR MULTI-STEP REASONING

We extend our nonconformity scoring function to operate at the trajectory level. The trajectory-level entailment operator is defined as:

$$\mathcal{E}_H(y^*) = \bigcap_{h=1}^{H} \mathcal{E}(y_h^*)$$

where $\mathcal{E}(y_h^*)$ is the standard entailment operator from Equation (1).

The trajectory-level nonconformity score becomes:

$$r_H(X_i, y_i^*) := \min \{\mathrm{rank}(S) \mid S \in R_i, \ y_i^* \notin \mathcal{E}_H(M(S))\} - 1$$

where $M(S)$ merges the entire reasoning trajectory $S$ into a coherent response.

For a new input $X_{n+1}$, our calibrated prediction function is:

$$L_\alpha^H(X_{n+1}) = M \left( \bigcup_{j:\mathrm{rank}(S^{(j)}) \leq \hat{q}_\alpha} S^{(j)} \right)$$

where $S^{(j)}$ denotes the $j$-th highest ranked reasoning trajectory.

**Theorem F.1** (Multi-step Coverage Guarantee). *Let $\{(X_i, y_i^*)\}_{i=1}^{n+1}$ be exchangeable, and let $\hat{q}_\alpha$ be the $\lceil (n+1)(1-\alpha) \rceil / n$-quantile of $\{r_H(X_i, y_i^*)\}_{i=1}^{n}$. Then:*

$$\mathbb{P}\left(y_{n+1}^* \in \mathcal{E}_H(M(L_\alpha^H(X_{n+1})))\right) \geq 1 - \alpha$$

*Proof.* The proof follows Theorem 4.1 by treating the entire reasoning trajectory as a single composite output. By the properties of conformal prediction:

$$\mathbb{P}(r_H(X_{n+1}, y_{n+1}^*) \leq \hat{q}_\alpha) \geq 1 - \alpha$$

If $r_H(X_{n+1}, y_{n+1}^*) \leq \hat{q}_\alpha$, then all trajectories $S^{(j)}$ with $\mathrm{rank}(S^{(j)}) \leq \hat{q}_\alpha$ are factually correct across all steps. Therefore:

$$\mathbb{P}\left(y_{n+1}^* \in \mathcal{E}_H(M(L_\alpha^H(X_{n+1})))\right) \geq 1 - \alpha$$

$\square$

To enhance conditional validity in multi-step scenarios, we extend AdaptiveRankConf by incorporating step-specific difficulty features $Z_{i,h}$ that capture step complexity, context dependency depth, and domain-specific indicators. The adaptive threshold function becomes trajectory-aware:

$$\hat{q}_\alpha(Z_{n+1}) = \sup\{r : r \le g_r(Z_{n+1})\}$$

where $Z_{n+1} = (Z_{n+1,1}, \ldots, Z_{n+1,H})$ and $g_r$ solves the optimization problem in Equation (16) with trajectory-level features. This extension maintains theoretical guarantees while improving empirical performance on complex reasoning tasks.

## G  RESULTS OF THE COMPARISON METHOD FOR SUBCLAIMS FACTUAL FILTERING

In this section, we present four specific case studies to illustrate the effectiveness of our approach in fact-checking after decomposing responses to subclaims. Based on the accuracy of factual implications, our method **RankConf** determines the correctness of subclaims more accurately than the comparative method SplitConf on problems without $Z_i^G$ group feature conditions. Similarly, our method **AdaptiveRCf** is also more accurate than the comparative method CondSplitConf on problems within-group $Z_i^G$ feature conditions.

Table 10: Subclaims filtering comparison for question "Find $x$ such that $\lceil x \rceil + x = \frac{23}{7}$. Express $x$ as a common fraction." from **MATH** dataset with setting (I-Model=gemini-2.0-flash, II-Model=gemini-2.5-pro, $\alpha = 0.1$).

| Subclaims $K = 10$ | Entailment | SplitConf | RankConf |
|---|---|---|---|
| The solution is $x = \frac{9}{7}$. | ✔ | ✔ | ✔ |
| Since $\lceil x \rceil$ must be an integer, let $\lceil x \rceil = 2$, then $x = \frac{23}{7} - 2 = \frac{9}{7}$. | ✔ | ✔ | ✔ |
| $x$ is less than $\lceil x \rceil$ and greater than $\lceil x \rceil - 1$, which is a fundamental property of ceiling functions. | ✔ | ✔ | ✔ |
| For this equation, $\lceil x \rceil = 2$ and $1 < x < 2$. | ✔ | ✘ | ✔ |
| The fraction $\frac{9}{7}$ is in simplest form with denominator 7. | ✔ | ✘ | ✔ |
| When solving $\lceil x \rceil + x = \frac{23}{7}$, we find that $x = 1.2857$, which equals $\frac{9}{7}$. | ✔ | ✔ | ✔ |
| Since $\frac{23}{7} \approx 3.286$, and $\lceil x \rceil$ must be 2, the solution is $x = \frac{9}{7}$. | ✔ | ✔ | ✔ |
| The ceiling function $\lceil x \rceil$ equals 3 for this problem, leading to $x = \frac{2}{7}$. | ✘ | ✘ | ✘ |
| If $x = \frac{16}{7}$, then $\lceil x \rceil + x = 3 + \frac{16}{7} = \frac{37}{7}$, which is not $\frac{23}{7}$. | ✘ | ✔ | ✘ |
| $x$ must be greater than 1.5 for the equation $\lceil x \rceil + x = \frac{23}{7}$ to hold true. | ✘ | ✘ | ✘ |

Table 11: Subclaims filtering comparison for question "A bookstore is deciding what price it should charge for a certain book. After research, the store finds that if the book's price is $p$ dollars (where $p \leq 32$), then the number of books sold per month is $128 - 4p$. What price should the store charge to maximize its revenue?" from **MATH** dataset with setting (I-Model=gemini-2.0-flash, II-Model=gemini-2.5-pro, $\alpha = 0.1$).

| Subclaims $K = 10$ | *Entailment* | SplitConf | RankConf |
|---|---|---|---|
| The revenue is calculated by price multiplied by quantity sold, which gives $R(p) = p(128 - 4p)$. | ✓ | ✓ | ✓ |
| The revenue function $R(p) = 128p - 4p^2$ is a quadratic function that opens downward. | ✓ | ✓ | ✓ |
| To maximize revenue, the store should charge \$16 per book, as this is the vertex of the parabola. | ✓ | ✓ | ✓ |
| At the optimal price of \$16, the store will sell 64 books per month. | ✓ | ✓ | ✓ |
| The maximum monthly revenue will be $1,024$ when the price is set to $16$. | ✓ | ✗ | ✓ |
| The derivative of the revenue function is $R'(p) = 128 - 8p$, and setting this equal to zero gives the optimal price. | ✓ | ✓ | ✓ |
| The price elasticity of demand at the revenue-maximizing price is exactly -1. | ✓ | ✓ | ✓ |
| If the price is set to $20$, the store will sell $48$ books and generate $960$ in revenue. | ✓ | ✓ | ✓ |
| The optimal price is \$24, which is three-fourths of the maximum allowable price of \$32. | ✗ | ✗ | ✗ |
| When the price exceeds \$25, the revenue begins to increase again. | ✗ | ✓ | ✗ |

Table 12: Subclaims filtering comparison for question "Are genital warts serious?" ($Z_i^G$ Group: level 1 difficulty) from **MedicalQA** dataset with setting (I-Model=gemini-2.0-flash, II-Model=gemini-2.5-pro, $\alpha = 0.1$).

| Subclaims $K = 10$ | *Entailment* | CondSConf | AdaptiveRCf |
|---|---|---|---|
| Genital warts are caused by certain strains of the human papillomavirus (HPV). | ✓ | ✓ | ✓ |
| Genital warts, caused by certain strains of the human papillomavirus (HPV), are a common sexually transmitted infection. | ✓ | ✓ | ✓ |
| While genital warts can be uncomfortable, they are not considered a serious health threat. | ✓ | ✓ | ✓ |
| The primary concern with genital warts is physical discomfort rather than serious health consequences. | ✓ | ✓ | ✓ |
| Genital warts are not life-threatening and can be effectively managed with proper treatment. | ✓ | ✗ | ✓ |
| Most cases of genital warts resolve on their own without causing significant health issues. | ✓ | ✗ | ✓ |
| Genital warts are considered highly dangerous and often lead to cancer if untreated. | ✗ | ✗ | ✗ |
| Genital warts can cause severe internal organ damage if left untreated. | ✗ | ✓ | ✗ |
| Genital warts often lead to severe immune system failure in real case. | ✗ | ✗ | ✗ |
| Genital warts are commonly prescribed antibiotics for treatment. | ✗ | ✗ | ✗ |

Table 13: Subclaims filtering comparison for question "What ingredient in walnut interferes with Synthroid drug absorption?" ($Z_i^G$ Group: level 2 difficulty) from **MedicalQA** dataset t with setting (I-Model=gemini-2.0-flash, II-Model=gemini-2.5-pro, $\alpha = 0.1$).

| Subclaims $K = 10$ | Entailment | SplitConf | AdaptiveRCF |
|---|---|---|---|
| The primary component in walnuts that interferes with Synthroid absorption is dietary fiber. | ✓ | ✓ | ✓ |
| Walnut consumption should be separated from thyroid medication intake by at least 4 hours to avoid absorption interference. | ✓ | ✓ | ✓ |
| Dietary fiber, particularly from sources like walnuts, can bind to levothyroxine (the active ingredient in Synthroid) and reduce its absorption in the gastrointestinal tract. | ✓ | ✗ | ✓ |
| The interference occurs due to the formation of insoluble complexes between dietary fiber and thyroid hormones in the digestive system. | ✓ | ✗ | ✓ |
| Clinical studies have shown that consuming walnuts with Synthroid can reduce drug absorption by approximately 20-30%. | ✓ | ✓ | ✓ |
| Patients taking thyroid medication should be advised to avoid consuming walnuts within several hours of taking their medication. | ✓ | ✓ | ✓ |
| Walnuts contain high levels of omega-3 fatty acids which directly inhibit the absorption of levothyroxine. | ✗ | ✗ | ✗ |
| The phytic acid content in walnuts forms complexes with Synthroid, preventing its intestinal absorption. | ✗ | ✓ | ✗ |
| Walnuts contain goitrogens that increase TSH production and counteract Synthroid's effects. | ✗ | ✗ | ✗ |
| The magnesium content in walnuts binds to thyroid medication, creating insoluble compounds that cannot be absorbed. | ✗ | ✗ | ✗ |

Table 14: Subclaims filtering comparison for question "What is prednisone used for?" ($Z_i^G$ Group: level 3 difficulty) from **MedicalQA** dataset t with setting (I-Model=gemini-2.0-flash, II-Model=gemini-2.5-pro, $\alpha = 0.1$).

| Subclaims $K = 10$ | Entailment | CondSConf | AdaptiveRCf |
|---|---|---|---|
| Prednisone is a corticosteroid medication used to suppress the immune system and reduce inflammation. | ✓ | ✓ | ✓ |
| Prednisone is primarily used as an anti-inflammatory and immunosuppressive agent. | ✓ | ✗ | ✓ |
| Prednisone is frequently prescribed for severe allergies and asthma. | ✓ | ✓ | ✓ |
| The primary purpose of prednisone is to reduce inflammation and suppress immune responses. | ✓ | ✗ | ✓ |
| Prednisone is often prescribed for its anti-inflammatory and immunosuppressant properties. | ✓ | ✗ | ✗ |
| Prednisone is primarily used as an antibiotic for bacterial infections. | ✗ | ✓ | ✗ |
| Prednisone can cure viral infections by enhancing immune response. | ✗ | ✗ | ✗ |
| Prednisone is commonly used as a painkiller for chronic headaches. | ✗ | ✓ | ✗ |
| Prednisone is used to increase blood pressure in patients with hypotension. | ✗ | ✗ | ✗ |
| Prednisone is primarily used to treat diabetes by regulating blood sugar levels. | ✗ | ✗ | ✗ |

# H    FURTHER DISCUSSION ON FUTURE WORK AND LIMITATIONS

While our framework demonstrates strong performance on factual question answering tasks with well-defined ground truths, we acknowledge several promising directions for future research that address current limitations.

**Extension to creative generation tasks.** A natural extension of our work would involve adapting conditional conformal prediction guarantees to creative generation scenarios such as storytelling, poetry composition, or multimodal (text-image) interpretation. However, this presents significant challenges as these domains typically lack objective factual labels and entailment relationships that form the foundation of our current framework. Unlike question answering where correctness can be evaluated against reference knowledge $y^*$ via the entailment operator $\mathcal{E}(\cdot)$ as defined in Equation (1), creative tasks require alternative evaluation paradigms based on subjective quality, coherence, or stylistic alignment (Quach et al., 2023; Silva-Rodríguez et al., 2025). Future work might explore hybrid approaches that combine human preference data with conformal guarantees, or develop domain-specific nonconformity scores based on stylistic features and coherence metrics rather than factual correctness.

**Toward integrated reasoning-time stopping mechanisms.** Another valuable direction involves integrating conformal prediction directly into the reasoning process of large language models to enable early stopping when reasoning errors are detected. Our method, like other mainstream approaches in this field, operates on fully generated responses rather than intermediate reasoning steps, and thus is not suitable for real-time intervention during the generation process (Jung et al., 2023; Detommaso et al., 2024). The method operates on fully generated responses rather than intermediate reasoning steps, making it unsuitable for real-time intervention during generation. To achieve true early stopping capabilities, future frameworks would need to develop conformal guarantees at intermediate reasoning stages, potentially through step-wise nonconformity scores or hierarchical threshold mechanisms that can identify reasoning errors before completion. Alternatively, develop a method for conformal prediction that can handle online streaming data (Zhang et al., 2025).

These directions represent important frontiers for conformal language generation. Such advances would substantially broaden the applicability of conformal prediction to diverse language generation scenarios beyond factual question answering, enabling reliable uncertainty quantification across the full spectrum of language model capabilities.

