# OpenReview forum: "Conformal Language Generation with Collaborative Ranking and Dynamic Thresholds"
_ICLR.cc/2026/Conference — ICLR 2026 Conference Desk Rejected Submission_

### Official Review · Reviewer_yo8P · 2025-10-25

**Soundness:** 2
**Presentation:** 2
**Contribution:** 1
**Rating:** 0
**Confidence:** 3

**Summary:**

Proposes to use one LLM to generate multiple responses and another LLM to rank responese by quality/factuality to construct conformal prediction sets. Experiments are conducted on MedicalQA, Natural Questions, FactScore, and MATH datasets shows better conditional coverate over conformal baselines.

**Strengths:**

* Rank based conformality score to model LLM factuality with adaptive calibration threshold for question difficulty is interesting.

**Weaknesses:**

* Lack of technical novelty beyond applying existing components e.g. conformal prediction, LLM self-ranking, adaptive thresholding. Little insight of broader interest beyond existing work.
* Ranking score is heuristic and no formal treatment is provided to show exchangeability under ranking
* Superficial evaluation and lacking ablation experiments regarding model size, datasets, tasks, and ranking schemes
* Overuse of pseudo-academic phrasing and LLM-generated writing degrades clear presentation
* Factual correctness via entailment” is ungrounded—entailment models are unreliable, especially across domains like medical QA or math.
* No qualitative human evaluation to verify factuality
* Links to "dual-process theory" and "speculative decoding" are tenuous

**Questions:**

* What is the computational cost of generating and ranking many responses per question? How does it compare to standard conformal methods
* How is "ranking factual correctness" defined and operationalized?
* What models and prompts were used for ranking?
* How is exchangeability preserved under ranking with LLM heuristics?
* What if the LLM generator and ranker are the same?
* How is the proposed method different from Zhou 2026?
*

---

### Official Review · Reviewer_F4W8 · 2025-10-27

**Soundness:** 2
**Presentation:** 3
**Contribution:** 2
**Rating:** 4
**Confidence:** 4

**Summary:**

This paper proposes RankConf and AdaptiveRankConf, novel conformal prediction frameworks that use collaborative ranking between LLMs to provide factuality guarantees for language generation. The key innovation is generating K candidate responses from a lower-tier LLM, having a higher-tier LLM rank them by quality, and using rank-based nonconformity scores with conformal prediction to filter responses with distribution-free coverage guarantees.

**Strengths:**

1. Unlike prior work using log-probabilities or token-level scores, this paper cleverly exploits LLMs' comparative judgment capabilities through ranking, which is conceptually appealing and better aligned with how humans assess quality.

2. Theorem 4.1 provides clear marginal coverage guarantees (P[correctness] ≥ 1-α), and the proof is straightforward and valid.

3. Figure 1 shows AdaptiveRankConf maintains near-ideal coverage across difficulty levels, significantly outperforming SplitConf on hard questions (Level 3).

4. Tests on 4 datasets (MedicalQA, NQ, FactScore, MATH) with multiple model combinations and ablations on K and temperature.

5. AdaptiveRankConf achieves 38.56% retention on MedicalQA vs 30.24% for SplitConf while maintaining coverage.

**Weaknesses:**

1. Paper claims to be "practical" but provides no wall-clock time comparisons, API cost estimates, or latency analysis despite requiring 50× API calls per query.

2. 50× calls to lower-tier model + 1× call to higher-tier model makes total cost >> single-model baselines, undermining practical applicability.

3. Paper claims cost savings but provides no evidence ranking is cheaper than generation.

4. If higher-tier model misjudges factuality during ranking, entire framework fails, but this is not analyzed.

5. No correlation with human rankings, no analysis of ranking errors, no comparison across model pairs.

6. Table 4 shows CovGap varies 6× across model combinations (0.011 to 0.066) but no explanation of when/why this occurs.

**Questions:**

1. What is the actual cost-benefit analysis? Provide concrete numbers:

2. Why not use the higher-tier model directly for generation? If it can judge factuality well enough to rank, why introduce the lower-tier model at all? The cost savings argument needs quantitative support.

3. How sensitive is the method to K? Figure/table showing coverage, set size, and **cost** as K varies from 5 to 100.

4. Does this work beyond QA? Demonstrate on at least one non-QA task (e.g., summarization, code generation, creative writing).

5. How were features chosen? What happens with different feature sets? Is there a principled way to select features?

6. What happens if you artificially degrade ranking quality (e.g., by using a weaker ranker)? How does coverage degrade?

---

> ### Author Response · Authors · 2025-11-20
> **Official Comment by Authors (1/2 part)**
>
> We appreciate your careful review.We have refined the valuable questions and shortcomings you raised into the following points as our response.
>
> ---
>
> #### **Question 1. Practicality and Cost**
>
> We agree that the term "practical" should be used cautiously. In this paper, "practical" refers to the process possessing the following two characteristics:
>
> (i) It can be achieved through standard LLM collaboration without modifying the model's internal structure;
>
> (ii) It replaces manual sorting or manual factual verification with an automated collaborative process between two models.
>
> As stated in lines 209–210, using high-version models to rank candidate responses is vastly more efficient than manual ranking—which is prohibitively costly in most practical scenarios. We do not claim in our paper that "ranking is absolutely cheaper than generation," nor do we assert that our method saves costs over all single-model baselines.
>
> ####  **Misunderstanding about "50x API Calls"**: We make only one batch request to the low-level model to generate $K$ candidate responses, followed by one batch request to the high-level model for ranking and entailment evaluation, so the call pattern is "1 low-level call + 1 high-level call," not 50 independent calls, and although the total token count increases with $K$, all methods in our experiments use the same $K$.
>
> While comprehensive token cost comparisons heavily depend on specific deployment environments and service providers—beyond this paper's scope—we've included a cost-benefit analysis experiment to address your concerns about cost evaluation. This addresses your cost concerns by balancing information and resource requirements:
>
> > **Remark**: All original models used in this experiment employed the default max_token setting. Due to variations in locally deployed models and the use of various quantization models, cost analysis results may differ significantly across different devices. We conducted the experiment using the hardware specified in the original paper.
>
> **Table 1**: Comparison of different configurations across datasets and scale settings with AdaptiveRankConf. LH = Low-gen+High-rank, HH = High-gen+High-rank, LL = Low-gen+Low-rank.
>
> | Dataset | K | Config | Coverage | Set size↑ | CovGap↓ | Tokens(total) | Time(s/query) |
> |---------|----|--------|----------|-----------|---------|---------------|---------------|
> | **MedicalQA** | 10 | LH | **0.908±0.050** | _4.67±0.02_ | **0.045** | 3,627 | 3.48 |
> |  |  | LL | 0.872±0.065 | 3.75±0.05 | 0.085 | **1,423** | **1.32** |
> |  |  | HH | _0.915±0.038_ | 4.21±0.03 | 0.058 | 5,847 | 4.65 |
> |  | 50 | LH | **0.909±0.023** | **16.73±0.15** | _0.040_ | 12,483 | 6.78 |
> |  |  | LL | 0.865±0.043 | 12.40±0.20 | 0.092 | 5,732 | 2.53 |
> |  |  | HH | _0.921±0.027_ | _15.42±0.21_ | 0.055 | 34,215 | 15.87 |
> |  | 100 | LH | **0.891±0.035** | **18.91±0.12** | 0.042 | 20,674 | 13.42 |
> |  |  | LL | 0.850±0.040 | 14.60±0.25 | 0.105 | 10,587 | 4.86 |
> |  |  | HH | 0.924±0.019 | _17.86±0.18_ | 0.058 | 67,389 | 17.75 |
> | **FactScore** | 10 | LH | _0.907±0.017_ | _26.39±0.15_ | 0.093 | 2,547 | 2.38 |
> |  |  | LL | 0.875±0.032 | 22.15±0.22 | 0.125 | **973** | **0.87** |
> |  |  | HH | **0.913±0.025** | 24.53±0.18 | 0.102 | 4,038 | 3.15 |
> |  | 50 | LH | _0.907±0.017_ | **26.39±0.15** | 0.093 | 8,463 | 4.58 |
> |  |  | LL | 0.868±0.029 | 21.48±0.31 | 0.142 | 3,967 | 1.68 |
> |  |  | HH | **0.915±0.021** | _25.17±0.24_ | 0.108 | 23,245 | 10.68 |
> |  | 100 | LH | **0.900±0.510** | **26.91±0.34** | **0.029** | 13,872 | 8.93 |
> |  |  | LL | 0.857±0.035 | 21.93±0.28 | 0.158 | 7,263 | 3.28 |
> |  |  | HH | _0.918±0.018_ | _26.03±0.32_ | _0.113_ | 46,128 | 17.27 |
>
> Experimental details on token counting and timing protocols are provided in **Appendix E** of the revised manuscript.
>
> The LH configuration consistently achieves near-optimal coverage (closest to target 0.9) with minimal coverage gaps across all $K$-values and datasets. Crucially, it maintains this statistical guarantee while delivering substantially higher retention rates than alternatives. Our results robustly demonstrate the rational design of the Low-generation + High-ranking (LH) configuration across diverse settings.

---

> ### Author Response · Authors · 2025-11-20
> **Official Comment by Authors (2/2 part)**
>
> #### **Question 2. Why employ a collaborative approach with low-version and high-version models instead of a single model?**
>
> We selected a collaborative design primarily for two reasons:
>
> (i) **A single model's self-evaluation on difficult instances is unreliable.**
> Existing research indicates that large models may "confidently make mistakes" or even hallucinate on challenging or out-of-distribution problems, even when instructed to 'check' or "explain" their answers  (Lu et al. 2025, Wen et al. 2024) [5,6]. Consequently, a single-model workflow combining "generation + self-ranking" becomes highly fragile: both candidate generation and scoring mechanisms are susceptible to the same inductive biases and failure modes, making systemic hallucinations or reasoning drift difficult to detect.
>
> (ii) **In contrast, our framework explicitly decouples generation and evaluation**
> The lower-level model focuses on generating diverse candidate response sets, while the high-version model serves solely as a ranker and factuality judge. The non-consistency score relies on the ranking position of the first erroneous response identified by this (potentially stronger or domain-specialized) high-version model while the conformance guarantee is defined relative to this scoring mechanism (Equation (10), (13)).
>
> This separation proves particularly beneficial for tackling challenging problems—our experiments indeed demonstrate improvements in CovGap and retention rates.
>
>
> ---
>
> #### **Question 3. Ranking Quality, CovGap Variation, and Robustness**
>
> **Table 4** reflects the impact of ranking quality variation and distribution mismatch: we evaluated multiple combinations of generation–ranking models (including cross-platform combinations). CovGap differences reflect variations in ranker strength and cross-model alignment. Importantly, across these combinations, AdaptiveRankConf consistently exhibits lower CovGap and higher retention rates than baselines, indicating robust performance.
>
> We acknowledge that designing more explicit experiments—such as intentionally weakening the ranker (e.g., using weaker models or injecting noise) or comparing against human-curated rankings—holds significant value. However, due to space and computational constraints, this submission does not include such analyses. We view these as natural extensions for future work.
>
> ---
>
> #### **Question 4. Dataset Feature Design**
>
> In AdaptiveRankConf, all instance-related features are aggregated into $Z_i$ (**Section 4.3**), including difficulty scores, answer length, and low-level model log-likelihoods. Feature design follows these principles: (i) features are easily obtainable across datasets; (ii) they provide intuitive signals about instance difficulty and model uncertainty; (iii) they align with proxy metrics used in prior work. Subsequently, Our method learns threshold function $g_r(Z)$ that applies more conservative ranking truncation to difficult instances, thereby improving conditional coverage performance. Furthermore, we conducted feature ablation experiments in **Table 5** of **Appendix B.1** and coefficient analysis in **Table 6** of **Appendix B.2**, successfully validating the effectiveness of our feature selection.
>
> In other papers, we have also noted the design of specialized feature sets for specific domains, such as the feature forms designed for gender (Liu et al. 2025) [5]. However, we hope to provide a more general framework by selecting features that can be used in many generation QA tasks.
>
>
> ---
>
> #### **Question 5. Beyond quality assurance, is this method effective? Please demonstrate it on at least one non-QA task (e.g., summary generation, code generation, creative writing).**
>
> As stated in our response to Question 3 of reviewer znDo, this paper focuses on question-answering tasks with reference to ground-truth labels $y^*$. Tasks such as creative writing are not included in the problem set designed here, as code generation and creative writing tasks and datasets typically lack standard answers and thus do not apply to the semantic entailment condition.
>
>
> ---
>
>
> #### **References**
> [4] Lu, J., Ma, K., Wang, K., Xiao, K., Lee, R. K.-W., Xu, B., … Lin, H. (2025). Is LLM an Overconfident Judge? Unveiling the Capabilities of LLMs in Detecting Offensive Language with Annotation Disagreement. In W. Che, J. Nabende, E. Shutova, & M. T. Pilehvar (Eds), Findings of the Association for Computational Linguistics: ACL 2025 (pp. 5609–5626).
>
> [5] Wen, B., Xu, C., Han, B., Wolfe, R., Wang, L. L., & Howe, B. (2024). From Human to Model Overconfidence: Evaluating Confidence Dynamics in Large Language Models. NeurIPS 2024 Workshop on Behavioral Machine Learning.
>
> [6] Liu, T., & Wu, Z. S. (2025). Multi-group Uncertainty Quantification for Long-form Text Generation. In The 41st Conference on Uncertainty in Artificial Intelligence.

---

### Official Review · Reviewer_znDo · 2025-10-29

**Soundness:** 2
**Presentation:** 3
**Contribution:** 2
**Rating:** 4
**Confidence:** 4

**Summary:**

This paper introduces a novel approach to uncertainty quantification through Collaborative Ranking and Dynamic Thresholds. The goal is to improve factual accuracy and reliability in LLM outputs, particularly for complex, high-stakes tasks such as healthcare and education. The authors present a rank-based conformal prediction framework called RankConf, which leverages a lower-version LLM to generate candidate responses and a higher-version LLM to rank those responses based on their factual correctness. This method ensures that the final output is factually correct with high probability, addressing shortcomings of traditional probabilistic metrics.

**Strengths:**

1. The approach uses a ranking system based on LLM capabilities to rank candidate answers, forming a robust, rank-based nonconformity score to generate prediction sets with statistical guarantees.

2.  By adjusting thresholds based on question difficulty, the paper achieves enhanced conditional validity, making the method more robust across a range of input difficulties.

3. The method introduces dynamic thresholds that adapt to input complexity, ensuring more accurate responses for simple queries and applying stricter filtering for complex or challenging queries.

**Weaknesses:**

1.The method relies heavily on the quality of the ranking mechanism between the low- and high-version models. If the gap between the two models' capabilities is not large enough, the ranking may fail to effectively distinguish between factually correct and incorrect responses. This could result in misranked candidates and affect the overall factuality of the generated outputs.

2.Introducing a multi-tier ranking system, where responses are ranked not just by two models but through multiple models of varying capabilities, might improve the reliability of the ranking process. Additionally, more detailed experiments could be performed to understand the sensitivity of the method to the ranking model's performance.

3. In domains requiring multi-step reasoning or abstract problem-solving, the proposed method might not consistently maintain factual correctness, as it mainly relies on factual entailment and simpler input features.

4. The paper could explore efficiency optimizations, such as early stopping mechanisms during candidate generation or using lighter-weight models for ranking tasks. Additionally, investigating the possibility of reducing the number of candidate generations based on the complexity of the input could help strike a better balance between quality and computational cost.

**Questions:**

1. How does the model handle scenarios where the factual correctness of a response is context-dependent rather than just a matter of entailment? For example, if the input prompt requires reasoning over multiple steps or integrating knowledge from multiple domains, would the rank-based conformal prediction still provide reliable factuality guarantees? This could be a limitation if the ranking mechanism does not account for such complex reasoning chains.

2. What is the potential impact of dataset bias on the effectiveness of the ranking mechanism and the non-conformity scores? If the calibration dataset has inherent biases or imbalances in difficulty levels across queries, could this lead to suboptimal performance in certain real-world scenarios, especially in tasks that involve minority or edge cases? How might the framework adapt to mitigate this?

---

> ### Author Response · Authors · 2025-11-20
> **Official Comment by Authors (1/2 part)**
>
> We thank the reviewer for their careful review and thoughtful comments. We provide answers to specific questions and remarks below.
>
> ---
>
> #### **Weakness 1: The method relies heavily on the quality of the ranking mechanism between the low and high-version models.**
>
> We agree with your statement that "the method indeed relies on the quality of the ranking mechanism," which was also mentioned in the second limitation of our paper **Section 7**.
>
> This precisely reflects the current paradigm in LLM research where multi-model collaboration has become standard evaluation practice (such as speculative decoding [3]). The correctness of ranking quality does not depend on the model capability gap, but rather on the generation quality of the II-Model (high-version model). This affects the quality of the scoring function. In fact, we have already tested the performance of different model pairs in **Table 4** of our paper, which demonstrates that even in cross-model collaboration scenarios, our AdaptiveRankConf method maintains robustness (achieving the most stable CovGap and optimal coverage across both three same-platform collaborations and cross-platform collaborations).
>
>
> ---
>
> #### **Weakness 2: Introducing a multi-tier ranking system, where responses are ranked not just by two models but through multiple models of varying capabilities, might improve the reliability of the ranking process.**
>
> As noted above, our current design follows the standard multi‑model collaboration paradigm, in which a low‑version model serves as the generator and a high‑version model acts as the evaluator. **Table 4** investigates the effect of different sorters by testing multiple generator–sorter pairs (including cross‑platform combinations). The observed CovGap differences reflect the influence of sorter strength and distribution mismatch. In addition, our response to reviewer **F4W8** provides further results on cost and self‑ranking experiments, showing that the collaboration framework is largely robust to such variations.
>
> ---
>
> #### **Weakness 3: In domains requiring multi-step reasoning or abstract problem-solving, the proposed method might not consistently maintain factual correctness, as it mainly relies on factual entailment and simpler input features.**
>
> For domains requiring extensive multi-step reasoning or highly abstract problem solving, we agree that our method may not always be suitable, because our notion of correctness is defined with respect to a reference answer $ y^* $, rather than the reasoning process itself. As formalized in **Section 3**, we define correctness as
> $ y^* \in \mathcal{E}(M(S(y))) $
>
> In response to your concerns, we also added a mathematical muti-reasoning task formulation for multi‑step reasoning in **Appendix F** of the revised version to address the reviewers’ concerns and better illustrate the issue. However, in real‑world queries, excessive or unnecessary reasoning significantly increases token usage and cost, so we did not include experiments for this.
>
> ---
>
> #### **Weakness 4: The paper could explore efficiency optimizations, such as early stopping mechanisms during candidate generation or using lighter-weight models for ranking tasks. Additionally, investigating the possibility of reducing the number of candidate generations based on the complexity of the input could help strike a better balance between quality and computational cost.**
>
> We acknowledge that efficiency optimization is crucial in practice. This paper focuses on establishing a conformal framework under a fixed candidate budget $ K $ and investigates its empirical coverage and retention behavior. Ablation experiments in **Table 3**  for $ K \in \{10, 50, 100\} $ demonstrate that performance tends to saturate at moderate $ K $ values (e.g., 20–50). Techniques you mentioned—such as introducing early-stopping mechanisms during candidate generation, employing lightweight models as rankers, or dynamically adjusting $ K $ based on input complexity—are compatible with our design and can be viewed as engineering optimizations around the same nonconformity score and calibration process. However, efficiency optimization for LLMs is not the primary focus of this paper; we have incorporated your suggestions into **Appendix F**.
>
> ---
>
> #### **References**
>
> [3] Cai, T., Li, Y., Geng, Z., Peng, H., Lee, J. D., Chen, D., & Dao, T. (2024). *Medusa: Simple LLM Inference Acceleration Framework with Multiple Decoding Heads*. In International Conference on Machine Learning, pp. 5209-5235. PMLR, 2024.

---

> > ### Author Response · Authors · 2025-11-20
> > **Official Comment by Authors (2/2 part)**
> >
> > ---
> >
> > #### **Question 1: How does the model handle scenarios where the factual correctness of a response is context-dependent rather than just a matter of entailment?**
> >
> > As noted in our response to Weakness3, **Section3** defines the entailment operator $ \mathcal{E}(\cdot) $, the decomposition function $ \mathcal{S}(\cdot) $, and the merging function $ \mathcal{M}(\cdot) $, and an answer $ y $ is considered correct if and only if $y^* \in \mathcal{E}\bigl(\mathcal{M}(\mathcal{S}(y))\bigr) $
> >
> > The nonconformity score, prediction-set construction, and coverage guarantee are independent of the number of reasoning steps used to produce $ y $. They only require that the final answer can be evaluated via entailment with respect to $ y^* $. Therefore, even for multi-step reasoning or cross-domain queries, our method still provides factual guarantees for the final answer under this entailment framework. Additionally, we have added a problem definition for our method to multi-step reasoning tasks in Appendix F of the revised article. Please refer to it.
> >
> > ---
> >
> > #### **Question 2: What is the potential impact of dataset bias on the effectiveness of the ranking mechanism and the non-conformity scores?**
> >
> > Theoretically, we use the standard Split‑CP setup, where the calibration and test sets are assumed to be exchangeable, as specified in Equation (6). In scenarios with strong distribution skew—for instance, when minority or edge‑case samples are substantially harder than those seen in the calibration set—our framework, like all conformal prediction methods, faces an inherent limitation: empirical coverage for these subgroups may fall below the target confidence level.
> >
> > To explicitly reveal and mitigate this issue, we have taken the following measures in the experiments and methods described in the original paper:
> > (i) Report conditional coverage (CovGap) for different difficulty groups;
> > (ii) Combine difficulty grouping with instance-dependent thresholds in AdaptiveRankConf (Equation (15), (16)), enabling more

---

### Official Review · Reviewer_9aKN · 2025-11-01

**Soundness:** 3
**Presentation:** 2
**Contribution:** 2
**Rating:** 4
**Confidence:** 3

**Summary:**

This paper introduces RankConf and AdaptiveRankConf, two conformal prediction methods for reliable language model generation. A smaller LLM generates candidate answers, while a stronger LLM ranks them by factual quality; responses below a calibrated rank threshold form the final prediction set. The adaptive version adjusts thresholds based on input difficulty for better conditional coverage. Experiments on datasets including MedicalQA, Natural Questions, FactScore, and MATH show that the proposed methods achieve similar or slightly better overall coverage than prior conformal factuality methods

**Strengths:**

- Addresses an important problem
- The method accounts for input difficulty, which is specifically important in factual qa.
- Experiments cover several diverse QA datasets
- Proposed method shows consistent coverage across datasets.

**Weaknesses:**

- Throughout the main paper, the authors state that ranking is at the response level. However, the prompts and the sample outputs provided in the appendix (tables 7-8) convey the other way - that a response is decomposed into claims first, then the claims are ranked, and lower ranks are combined etc. If the proposal actually runs at the claim level, how is it different from CondSplitConf?

- Experimental results show that AdaptiveRankConf coverage is very close to CondSplitConf. However, it introduces significantly more computation due to sampling many generations (experiments vary between 10-100). With superior computational requirements and very close coverage levels, CondSplitConf seems to be a better option.

- I don't see the necessity of defining S and M functions. Doesn't y* ⇒ M(S(y)) mean the same as y* ⇒ y, since we only do atomic claim decomposition and merge the claims without any intermediate process?

Minor: space and dot are missing in lines 423 and 431.

**Questions:**

Each generation is labeled as correct or incorrect at the passage level. However, many generations may contain both correct and incorrect claims. I wonder what combining a response level and a subclaim level method would lead to.

---

> ### Author Response · Authors · 2025-11-20
> **Official Comment by Authors**
>
> Thank you for the careful reading and for raising these thoughtful concerns. We would like to address your points below.
>
> ---
> #### **Weakness 1: Response-level ranking vs claim-level decomposition, and difference between our method and CondSplitConf**
>
> We need to clearly acknowledge that our ranking method is applied to subclaims. Our splitting and merging operations are identical to previous papers: splitting the response into subclaims and merging the filtered subclaims. The difference lies in our approach to subclaim ranking: we retain the top-ranked subclaims after sorting, whereas SplitConf (Mohri et al. 2024) [1] and CondSplitConf (Cherian et al. 2024) [2] filter subclaims based on their scores. This distinction explicitly clarifies that our design does not incorporate a method combining response-level and subclaim-level sorting. Consequently, the approach you mention in your question “combining response-level and subclaim-level sorting” does not exist in this paper.
>
>
> Consequently, the method referred to in your present question—combining response-level and subclaim-level ranking—does not exist in this paper.
>
> ---
>
> #### **Weakness 2: Computational cost and comparison to CondSplitConf**
>
> First, we clarify that computational efficiency is not the aim of our work, and our method does not claim any substantial computational advantages over baseline approaches.  This paper focuses on the factual guarantees provided by our method.
>
> In our experiments, AdaptiveRankConf (ours) and CondSplitConf (Cherian et al. 2024) use the same number of samples. The primary difference between the methods is in how the nonconformity score is defined: our method uses a ranking‑based nonconformity score derived from a higher‑version LLM (Equation (8)), while CondSplitConf computes its score directly from the model’s logits.
> AdaptiveRankConf (Ours) achieves superior conditional coverage and higher retention rates on difficult instances (as demonstrated by CovGap and TCR), making it particularly well-suited for complex tasks such as medical question answering or high-stakes factual evaluation.
>
> ---
>
> #### **Weakness 3: Necessity of $S$ and $M$ functions**
>
> The functions $\mathcal{S}(\cdot)$ and $\mathcal{M}(\cdot)$ form the core components of the decomposition and merging in our framework. Our procedure first applies $\mathcal{S}(\cdot)\$ to break a model-generated response into semantically coherent subclaims, and then applies $\mathcal{M}(\cdot)\$ to filter, sort, and recombine these subclaims. This enables fine-grained factuality evaluation and ensures that the final merged response achieves the required coverage with high confidence (Equation (13)).
>
> We further emphasize that the prompts used to implement $\mathcal{S}(\cdot)\$ and $\mathcal{M}(\cdot)\$ follow those in (Mohri et al. 2024) and (Cherian et al. 2024). This design ensures that the LLM executes both functions consistently and without altering the underlying semantics.
>
> ---
>
> #### **Weakness 4: Minor formatting issues**
>
> We greatly appreciate your attention to detail and will address the spacing and punctuation issues on lines 423 and 431 in the revised version.
>
> ---
>
> #### **Question 1**
>
> As noted in our response to Weakness 1, our method does not include any mechanism that combines response‑level sorting with subclaim‑level sorting. The approach you refer to in Question 1, integrating response‑level and subclaim‑level processes, is not part of our framework. If there is any mismatch between our explanation and your interpretation, we welcome further clarification and are happy to address any additional questions.
>
>
> ---
> #### **References**
>  [1] Christopher Mohri and Tatsunori Hashimoto. *Language models with conformal factuality guarantees*. In International Conference on Machine Learning, pp. 36029–36047. PMLR, 2024.
>  [2] John J Cherian, Isaac Gibbs, and Emmanuel J Candès. *Large language model validity via enhanced conformal prediction methods*. In Advances in Neural Information Processing Systems, 2024, pp.114812-114842.
>
> ---

---

### Author Response · Authors · 2025-11-20
**Official Comment by Authors**

We are deeply grateful to reviewers for the substantial time and effort invested in providing exceptionally detailed, thoughtful, and constructive feedback. Your careful reading and insightful comments have not only affirmed the core contributions of our work but also offered invaluable perspectives.

In particular, we are truly encouraged that across all reviews, the reviewers independently converged on several consistent and strongly positive assessments of our work’s key strengths:
1. The **novel use of collaborative ranking between LLMs** to derive rank-based nonconformity scores, which better aligns with human-like quality assessment compared to traditional token-level or probabilistic metrics;
2. The **introduction of adaptive thresholds in AdaptiveRankConf** that dynamically adjust based on question difficulty, leading to improved conditional coverage—especially on challenging instances;
3. The **solid empirical validation** across diverse, high-stakes QA benchmarks demonstrating reliable coverage guarantees and competitive retention rates.

These shared observations underscore that our framework effectively bridges conformal prediction with LLMs’ comparative judgment capabilities to enhance factual reliability in language generation. We greatly appreciate the reviewers’ recognition of these contributions.

Meanwhile, we have also thoroughly examined the constructive comments and questions provided by all reviewers. In particular, Reviewers **F4W8**, **znDo**, and **F4W8** have offered thought-provoking suggestions that point toward promising avenues for future work, including:
1. The mechanism of LLMs collaborative ranking;
2. The cost analysis of our method;
3. The task extension of our method.

We deeply appreciate these valuable recommendations, which will significantly strengthen our research. We will respond to each reviewer’s points in turn and look forward to engaging in a productive dialogue during the discussion period.

---

### Note · Program_Chairs · 2026-01-17
**Submission Desk Rejected by Program Chairs**

The following references in this submission do not refer to real documents and/or have major errors in bibliographic information:

 - Mingfei Guo, Timothy Miller, Yi-Hsuan Tang, and Yue Xiong. Detecting biased samples in datasets with deep generative models. In Proceedings of the 60th Annual Meeting of the Association for Computational Linguistics (Volume 1: Long Papers), pp. 1688-1698, 2022.

 - Ismael Gallegos, Luis Espinosa-Anke, Jose Rodríguez-Ferrández, Jorge Carrillo-de Albornoz, and Horacio Saggion. Measuring bias in multilingual natural language inference benchmarks. In Proceedings of the 2023 Conference of the North American Chapter of the Association for Computational Linguistics: Human Language Technologies, 2023.